# Lag-invariant detection of interactions in spatially-extended systems using linear inverse modeling

**Rikkert Hindriks**●*

Department of Mathematics, VU University Amsterdam, Amsterdam, The Netherlands

* r.hindriks@vu.nl

## Abstract

Measurements on physical systems result from the systems' activity being converted into sensor measurements by a forward model. In a number of cases, inversion of the forward model is extremely sensitive to perturbations such as sensor noise or numerical errors in the forward model. Regularization is then required, which introduces bias in the reconstruction of the systems' activity. One domain in which this is particularly problematic is the reconstruction of interactions in spatially-extended complex systems such as the human brain. Brain interactions can be reconstructed from non-invasive measurements such as electro-encephalography (EEG) or magnetoencephalography (MEG), whose forward models are linear and instantaneous, but have large null-spaces and high condition numbers. This leads to incomplete unmixing of the forward models and hence to spurious interactions. This motivated the development of interaction measures that are exclusively sensitive to lagged, i.e. delayed interactions. The drawback of such measures is that they only detect interactions that have sufficiently large lags and this introduces bias in reconstructed brain networks. We introduce three estimators for linear interactions in spatially-extended systems that are uniformly sensitive to all lags. We derive some basic properties of and relationships between the estimators and evaluate their performance using numerical simulations from a simple benchmark model.

## Introduction

Global patterns in complex systems emerge through interactions between large numbers of units. One way of characterizing the behavior of such systems is by applying network theory to the matrix of pair-wise interactions [1]. If inversion of the forward model is ill-posed, this approach leads to bias in the estimated interaction matrix and hence to distortions of reconstructed network topology [2]. Bias in estimated interaction matrices is well-known in the field of MEG and EEG and prevents a clear view on the functional organization of the human brain. Another example are local field potentials (LFP's), which are used in invasive neuro-physiological studies [3]. Although less ill-posed than the EEG/MEG inverse problems, the LFP inverse problem still requires regularization and this leads to bias in the reconstruction of

**Data Availability Statement:** All relevant data are within the paper and Supporting information files.

**Funding:** R.H. Rikkert Hindriks is supported by the NWO-Wiskundeclusters grant nr. 613.009.105, made available by the Nederlandse Organisatie

voor Wetenschappelijk Onderzoek (NWO) www.
nwo.nl. The funders had no role in study design,
data collection and analysis, decision to publish, or
preparation of the manuscript.

**Competing interests:** The authors have declared
that no competing interests exist.

current generators in neural tissue [4]. Although linear, these operators have large null-spaces
and are very sensitive to small perturbations, as evidenced by the high condition numbers of
their discretizations. In the current study, we propose a method to reduce the bias in estimated
interaction matrices that can be applied to observations obtained from arbitrary linear forward
models.

Estimation of system interactions from indirect sensor measurements is particularly well-
developed in the field of EEG and MEG, which are non-invasive techniques for detecting mag-
netic fields outside the head (MEG) and electric potentials on the scalp (EEG) set-up by cur-
rent generators in activated brain tissue [5]. In contrast to blood-oxygen level-dependent
(BOLD) functional magnetic resonance imaging (fMRI), MEG and EEG provide direct mea-
sures of neural activation with high temporal resolution and, as such, are indispensable tools
in fundamental and clinical human neuroscience. Although direct analysis of MEG and EEG
sensor data can be useful in disentangling cognitive processes, seizure detection, and classifica-
tion of pathologies, it does not provide insight into the spatiotemporal patterns of neural acti-
vation underlying the sensor data. This applies in particular to functional interaction analysis
[6, 7], which aims to characterize the temporal relationship between active sources [8], and is
therefore commonly carried out in source-space [6, 9]. Source-space interaction analysis, how-
ever, generally yields biased estimators due to residual mixing of the reconstructed source sig-
nals. This undesirable effect is referred to as *field spread* in the case of MEG and *volume-
conduction* in the case of EEG and is jointly referred to as *signal leakage*. Signal leakage is the
main obstacle in the non-invasive study of human brain interactions and has attracted consid-
erable attention from the research community.

Methods of dealing with signal leakage in source-space interaction analysis can roughly be
divided into two categories, corresponding to the type of interaction that is considered: signal
or amplitude interactions. *Signal interactions* refer to interactions between reconstructed
source-signals proper, whereas *amplitude interactions* refer to interactions between the signals'
amplitude fluctuations. Signal interactions characterize interactions between fast and usually
oscillatory signals on short time-scales, typically a fraction of the signals' oscillation period.
Examples are the Pearson correlation coefficient, coherence, and phase-synchronization [10].
Amplitude interactions characterize interactions between slowly varying amplitude envelopes
on time-scales in the BOLD-fMRI range (0.01-0.1 Hz). Methods of reducing leakage in signal
interactions are based on the observation that leakage is instantaneous and thus cannot explain
the observation of interactions with non-zero lag. Methods of this type therefore quantify
lagged interactions (i.e. interactions with non-zero lag) and discard all zero-lag interactions
[11–15]. Methods of reducing leakage in amplitude interactions remove all linear instanta-
neous interactions between the reconstructed source signals, before computing their ampli-
tude envelopes. This is done by applying a whitening transformation to the signals and is
referred to as *signal orthogonalization*. The amplitude envelopes are subsequently low-pass fil-
tered and functional interactions are quantified by the Pearson correlation coefficient [16–19].
This approach has enabled the discovery of the electrophysiological basis of human hemody-
namic resting-state networks [20, 21].

This study is concerned with signal interactions. Although the above mentioned methods
have proven valuable in various applications, their sensitivity decreases with decreasing (abso-
lute) lag, which leads to false negatives and distortions in the topology of reconstructed func-
tional networks [2]. This is a serious limitation, especially given that intra-cortical
electrophysiological signals are known to interact with small lags under many experimental
conditions and have been hypothesized to be critical for effective neuronal communication
and routing of information [22, 23]. The main reason why measures of signal interactions are
unable to separate true from spurious zero-lag interactions is that the formal relationship

between signal leakage, and the forward and inverse operators is not exploited. In [24, 25] this relationship is exploited for linear inverse operators by linking signal leakage to the algebraic structure of the resolution operator. This method, however, is designed for seed-based interactions and is asymmetric, without having a causal interpretation, which complicates its interpretation. Furthermore, straightforward symmetrization and generalization to the multivariate case leads to a variation on an earlier proposed bias-correction scheme that has been shown to yield only marginal improvements [26]. In [27], a subspace projection method is proposed for suppressing the contribution of source-power to the sensor-space spectral matrix. This yields a bias-corrected spectral matrix, which is subsequently projected to source-space using a non-linear scanning method. The method is based on the observation that contributions to the sensor-space spectral matrix that are due to source-power (spurious interactions) and source coherence (true interactions) lie in different linear subspaces of the vectorspace of $n \times n$ complex-valued matrices, where $n$ denotes the number of sensors. It is in this last step that the relationship between signal leakage and the forward matrix is exploited. The source-space spectral matrix is subsequently estimated by carrying out a non-linear scanning procedure in which the corrected sensor-space spectral matrix is compared with the (normalized) cross-product of leadfields of every pair of source locations. The main limitation of this method is the use of a non-linear scanning method, as such methods are known to be effective only for a small number of active sources [28].

The bias-reduction method proposed in [27] can in principle be combined with any inverse operator. This was noted in [27] but not worked out. In the current study we combine the bias-correction proposed in [27] with an arbitrary linear inverse operator to obtain a lag-independent estimator of system interactions that can be applied to reconstruct interactions in distributed source-activity. This requires to invert a linear forward model defined on vectorized matrix spaces. We also propose two novel bias-corrected estimators. The first estimator is obtained by interchanging the order of inversion and bias-correction. Thus, instead of correcting the sensor-space spectral matrix and subsequently projecting the corrected matrix to source-space, the sensor-data is first projected to source-space to yield reconstructed system activity, from which the spectral matrix can be estimated and corrected using a subspace projection in source-space. The second estimator is obtained by first solving the same vectorized inverse problem as above and subsequently correcting the reconstructed spectral matrix using the same source-space projection as above. This last estimator hence combines the ingredients of the other two estimators.

In **Background** we describe the linear forward model and linear inverse operator used in this study (***Linear inverse modeling***), define system interactions in the time-frequency domain using spectral matrices and formulate null-hypotheses and the construction of confidence regions (***Detection of system interactions***), introduce the benchmark model that will be used in evaluating the proposed methodology (***Benchmark model***), and motivate the use of complex-valued test-statistics for lag-independent detection of system interactions (***Test-statistics***). In **Estimation of system interactions** we describe uncorrected source- and sensor-space based estimators of system interactions and derive their filtering properties. In **The geometry of signal leakage** we provide a geometric characterization signal leakage in linear estimators of system interactions and provide some analytical examples using simple source configurations. In **Suppression of signal leakage** we use this characterization to construct three corrected interaction estimators and describe how their efficacy can be quantified. In **Comparative performance** we compare the performance of the corrected estimators with respect to suppression efficacy (***Effectiveness***), bias-reduction (***Bias reduction***), and detection power (***Detection power***).

## Background

### Linear inverse modeling

In the time-frequency domain, the electromagnetic forward model can be represented by a linear

$$y(t, \omega) = Lx(t, \omega) + e(t, \omega), \tag{1}$$

where $x(t, \omega)$ is a $p$-dimensional vector of brain activity, $y(t, \omega)$ is an $n$-dimensional vector of sensor measurements, $L$ is the forward matrix, $e(t, \omega)$ is an $n$-dimensional vector of measurement noise, and $t$ and $\omega$ denote time and angular frequency, respectively. A model of this form describes the relation between brain activity in the form of current source density (CSD) at $p$ locations within the brain and induced measurements at $n$ sensors, which can be electric potentials on the scalp as in electroencephalography (EEG), on the cortical surface as in electrocorticography (ECoG), or inside the brain as in local field potentials (LFP) recordings, as well as magnetic flux outside the head as in magnetoencephalography (MEG). These different types of measurement correspond to different forward matrices, which contain the relevant geometric and conduction properties of the volume conductor under study (i.e. the head) and are obtained by discretizing and solving the quasi-static approximation to Maxwells' equations [5]. Electromagnetic inverse modeling refers to the reconstruction of the CSD or any derived quantities from observed sensor measurements. What makes inverse modeling challenging is the fact that $p$ is generally much larger than $n$.

A popular way of obtaining a reconstruction $x^\star(t, \omega)$ of $x(t, \omega)$ is by solving the following penalized least-squares problem:

$$x^\star(t, \omega) = \arg\min_x \|y(t, \omega) - Lx(t, \omega)\|^2 + \lambda \|x(t, \omega)\|^2,$$

where $\lambda \geq 0$ is the noise regularization parameter [5, 28]. This problem has the unique solution

$$x^\star(t, \omega) = L^\sharp_\lambda y(t, \omega),$$

where the *inverse operator* $L^\sharp_\lambda$ is given by

$$L^\sharp_\lambda = L^T [LL^T + \lambda I]^{-1}. \tag{2}$$

In the neuroimaging community, $x^\star(t, \omega)$ is referred to as the *minimum norm reconstruction* of $x(t, \omega)$ and several generalizations of it have been developed [28]. The properties of a linear inverse operator are summarized by its *resolution matrix*. Given a forward matrix $L$ and an inverse operator $L^\sharp$, the *resolution operator* associated with $L$ and $L^\sharp_\lambda$ is defined as

$$R_\lambda = L^\sharp_\lambda L.$$

Its columns are known as *point-spread functions* and its rows is *cross-talk functions* [29, 30]. Note that in case of a single active source of unit strength at the $j$-th location, the reconstruction is equal to the $j$-th point-spread function. We will need these notions later on when analyzing the functional connectivity estimators.

### Detection of system interactions

We focus on the detection of interactions from ongoing recordings of the system's activity i.e. without perturbing the system. In neuroscience, this corresponds to recording brain activity in the absence of sensory stimulation and cognitive tasks. The proposed methodology, however,

can easily be adapted to detect interactions from recorded system responses to perturbations. We assume that ongoing system activity can be reasonably well modeled by a Gaussian stochastic process. Without loss of generality, we assume the process to have expectation zero. Such a process is completely determined by its *spectral matrix* $S_x(\omega)$, which is a function of (angular) frequency $\omega$ and is defined as

$$S_x(\omega) = \langle x(t,\omega)x(t,\omega)^\dagger \rangle,$$

where the brackets denote taking the expectation over time and the superscript † denotes taking the conjugate-transpose. The spectral matrix of a process is positive semi-define and conjugate symmetric, i.e. $S_x(\omega)^\dagger = \bar{S}_x(\omega)$, where the bar denotes entry-wise complex-conjugation and can hence be thought of as a covariance matrix in the time-frequency domain. Measurement noise is also modeled by a Gaussian stochastic process and for simplicity we will assume that $S_e(\omega) = \sigma^2 I$, where $I$ denotes the identity matrix i.e. the measurement noise is white, has the variance at each sensor, and is uncorrelated across sensors. The observed sensor data $y(t, \omega)$ therefore is also a stochastic process and has a spectral matrix $S_y(\omega)$ given by

$$S_y(\omega) = LS_x(\omega)L^T + \sigma^2 I.$$

In neuroscience, system interactions are referred to as *functional connectivity*, which is a broad term that encompasses notions based on different frameworks such as stochastic processes, weakly-coupled oscillators, chaos theory, and information theory [10]. We will adopt the framework of stochastic processes and hence characterize system interactions by the (non-diagonal entries of) the spectral matrix $S_x(\omega)$. Thus, the interactions between system locations $k$ and $l$ at frequency $\omega$ is characterized by the $(k, l)$-th entry of $S_x(\omega)$, which we will denote by $\gamma_{k,l}(\omega)$ and is a complex number whose magnitude and phase measure, respectively, the strength and lag (i.e. latency) of the interaction at frequency $\omega$.

This study is concerned with the construction of test-statistics for detecting the presence of system interactions based on observed sensor data. This problem is formalized by the following null-hypothesis:

$$H_0 : \gamma_{kl}(\omega) = 0,$$

against the two-sided alternative $H_1$: $\gamma_{kl}(\omega) \neq 0$, for all pairs $(k, l)$. The test-statistics we consider are complex-valued estimators of $\gamma_{k,l}$, together with their real and imaginary parts and are described in detail in **Estimation of system interactions** and **Suppression of signal leakage**. The performance of the test-statistics will be evaluated by computing the distance between $H_0$ and $100(1 - \alpha)$% confidence regions. Throughout the text we will refer to this distance as the *sensitivity* of a test-statistic and to the matrix carrying the sensitivities of all pairs $(k, l)$ as the *sensitivity matrix*. For complex test-statistics, $100(1 - \alpha)$% ellipsoidal confidence regions in the complex-plane are constructed and for its real and imaginary parts, $100(1 - \alpha)$% confidence intervals are constructed by simulating the statistics' sample distribution and calculating the $100(1 - \alpha/2)$% and $100\,\alpha/2$% percentiles. To compare the sensitivity of a complex test-statistic to that attained by the combined use of its real and imaginary parts, we set $\alpha = 0.05$ for the complex test-statistic and $\alpha = 0.025$ for its real and imaginary parts. This is done to correct for the fact that in the latter case, two tests need to be performed instead of one. The sample distributions of the different test-statistics are approximated by generating $10^3$ realizations of the simulated sensor data. In practice, these distributions can be approximated using appropriate bootstrap schemes.

## Benchmark model

To illustrate the methodology, we use a simple forward model in which electric potentials are measured that are induced by a current source density (CSD). The electric potential $V$ generated by a current source density $\rho$ in an infinite, homogeneous, and isotropic volume conductor is described by Poisson's equation:

$$\Delta V = \rho,$$

where $\Delta$ denotes the Laplace operator and where, without loss of generality, we have set the electrical conductivity to 1. In a neurophysiological context, $V$ is the local field potential (LFP) inside nervous tissue that is generated by transmembrane currents which are described macroscopically by the current source density $\rho$ [31, 32]. We consider a one-dimensional source-space segment of length 4 mm and a one-dimensional array of 11 electrodes with an inter-electrode spacing of 0.4 mm and that is located parallel to the source-space segment at a height of $h = 0.5$ mm (see Fig 1A).

The electric potential at location $y$ that is generated by a current monopole $\rho(x') = \delta(x' - x)$ of unit strength at location $x$ is given by the free-space fundamental solution to Poisson's equation:

$$V(x, y) = \frac{1}{4\pi\sqrt{h^2 + (x - y)^2}}.$$

The CSD is modeled by two monopoles located at locations -1 and 1 mm and correspond to the red dots in Fig 1A. The source-space is discretized using a spacing of 0.1 mm and hence gives rise to a $10 \times 81$ forward matrix $L$ that describes the relation between the (unobserved) CSD and the (observed) electric potentials. The $(k, l)$-th entry of $L$ is $V(x_l, y_k)$, where $x_l$ and $y_k$ denote the locations of the $k$-th electrode and the $l$-th source-point, respectively. The forward model has the form of Eq (1).

The two active sources are modeled by stationary oscillatory stochastic processes with spectral matrix

$$\begin{pmatrix} 1 & \gamma e^{i\phi} \\ \gamma e^{-i\phi} & 1 \end{pmatrix},$$

where $0 \leq \gamma \leq 1$ is the interaction strength (i.e. coherence) and $0 \leq \phi < 2\pi$ is the lag. Thus, the sources have the same amplitude and interact with strength $\gamma$ and a latency that is a fraction $\phi/2\pi$ of their average oscillation period. Measurement noise with spectral matrix $S_e = \sigma^2 I$ is added, where $\sigma^2$ is measured in units of the maximal eigenvalue of $LL^T$. Table 1 lists the model parameters and their nominal values. Fig 1B shows the observed spectral matrix $S_y$ that is generated by source activity with $\phi = 0$, $\gamma = 0$, and $\sigma = 0.1$. Note that although the two active sources can be discerned, due to the mixing of source activity by the forward model (two black curves), it is not immediately clear if, and to what extent, the sources are interacting.

## Test-statistics

A commonly used statistic in neuroscience for testing the null hypothesis $H_0$: $\gamma_{k,l} = 0$ of the absence of interactions between brain locations $k$ and $l$ is the imaginary part

$$T_{\text{imag}} = \text{Im}(\hat{\gamma}_{k,l}) = |\gamma_{k,l}| \sin(\hat{\phi}_{k,l}),$$

of some estimator $\hat{\gamma}_{k,l}$ of $\gamma_{k,l}$. Fig 2A shows $T_{\text{imag}}$ as a function of lag $\hat{\phi}_{k,l}$. It shows that $T_{\text{imag}}$ is

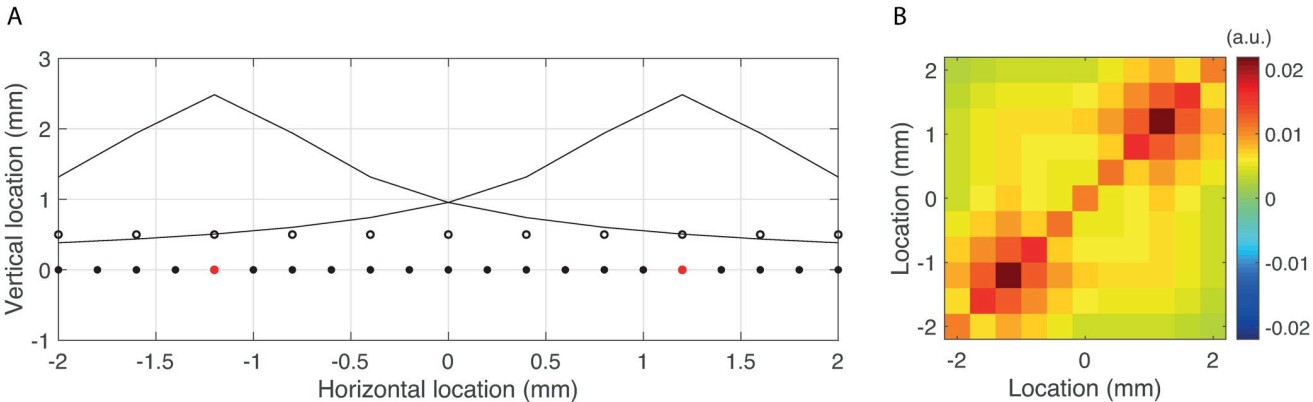

**Fig 1. Set-up of the test model.** A. Source-space segment of length 4 mm (horizontal axis at height 0 mm), recording electrodes (black dots at height 0.5 mm), and the locations of two active sources of neuronal activity (red dots). Also shown are the sensitivities of the electrodes to activity of the two sources (two black curves). B. Observed spectral matrix of the recorded electric potentials that are induced by non-interacting sources of unit strength and in the presence of measurement noise ($\sigma = 0.1$).

highest for imaginary interactions i.e. when system activity at locations $k$ and $l$ is coherent with lag one-fourth of the average oscillation cycle, and decreases to zero for purely real (i.e. instantaneous) interactions. Similarly, the sensitivity of the real part

$$T_{\text{real}} = \text{Re}(\hat{\gamma}_{k,l}) = |\gamma_{k,l}| \cos(\hat{\phi}_{k,l}),$$

is highest for instantaneous connectivity and decreases to zero for purely imaginary interactions (see Fig 2A). As a consequence, interactions with small/large lags might remain undetected. In neuroscience this leads to bias in measures that are derived from estimated interaction matrices such as network-theoretical measures that are often used to characterize the brain's functional organization [1, 33, 34].

To illustrate the dependence of the real and imaginary test-statistics on interaction lag, we computed their sensitivity as a function of lag, for $\hat{\gamma}_{k_0,l_0}$ being the $(k_0, l_0)$-th entry of the sample spectral matrix of the reconstructed source activity and $k_0$ and $l_0$ denote the true source locations. The number of samples was set to $N = 50$, the correlation level was set to $\gamma_{k,l} = 0.3$, and no measurement noise was added. Source activity was reconstructed using the inverse operator $L_\lambda^\sharp$ (see Eq (2)) where $\lambda$ was set to a small value ($\lambda = 10^{-20}$). All other parameters were chosen as in Table 1. The sensitivities are shown in Fig 2B. Observe that the sensitivity of $T_{\text{imag}}$ is zero for lags smaller than about 45 degrees, which means that interactions at lags less than 45

**Table 1. Model parameters, their symbols, and nominal values/ranges.**

| Parameter | Unit | Value |
|---|---|---|
| Array length | mm | 4.0 |
| Segment length | mm | 4.0 |
| Intra-sensor distance | mm | 0.4 |
| Intra-source distance | mm | 0.2 |
| Height of array ($h$) | mm | 0.5 |
| Interaction strength ($\gamma$) | - | 0.3 |
| Interaction lag ($\phi$) | deg | 0-90 |
| Noise level ($\sigma$) | - | 0-0.1 |

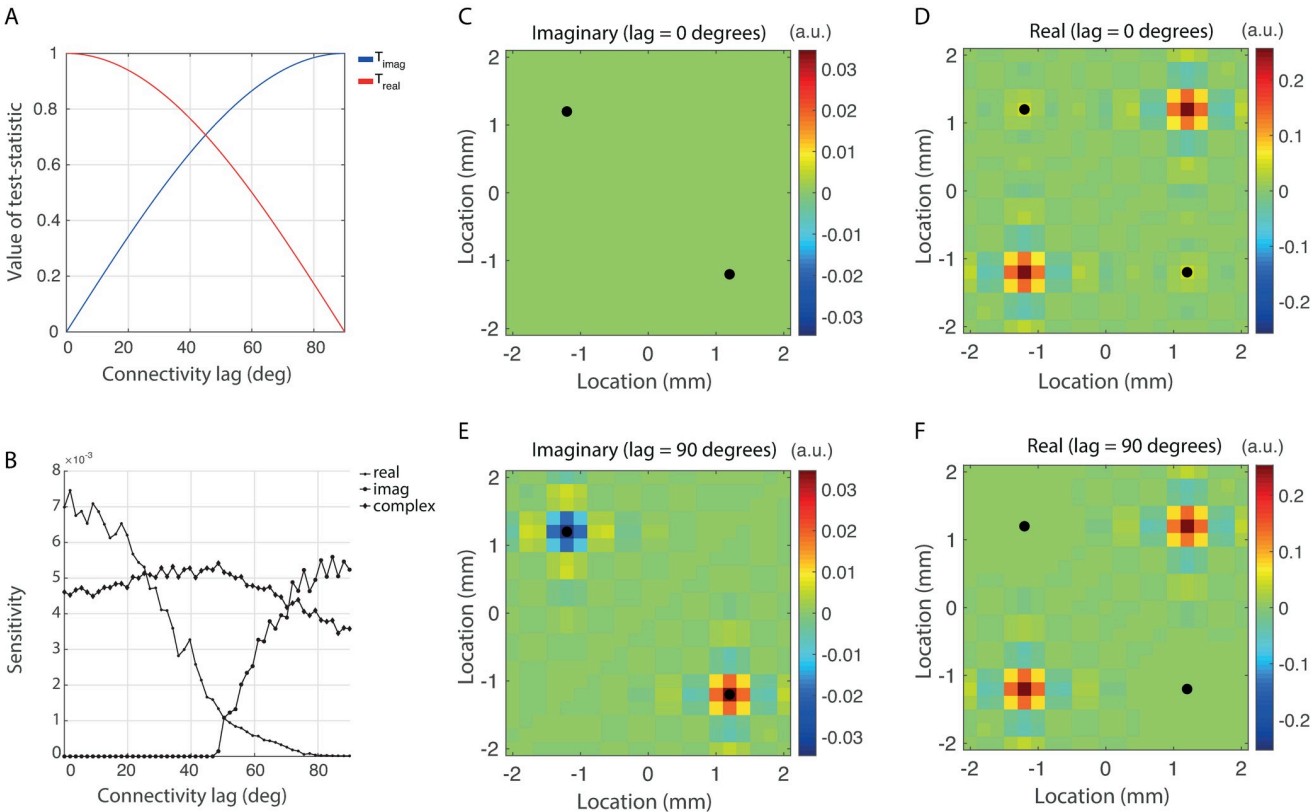

**Fig 2. Real, imaginary, and complex test-statistics.** A. Real-and imaginary part of the imaginary test-statistic as a function of lag. The strength of interaction was set to $\gamma_{k,l} = 1$ so that the curves range between zero and one. B. Sensitivity of the real, imaginary, and complex test-statistics at the true interaction-pair $(k_0, l_0)$, as a function of lag C. Sensitivity of the imaginary test-statistic for all interaction-pairs and for a lag of 0 degrees. D. Sensitivity of the real test-statistic for all interaction-pairs and for a lag of 0 degrees. E. Sensitivity of the imaginary test-statistic for all interaction-pairs and for a lag of 90 degrees. F. Sensitivity of the real test-statistic for all interaction-pairs and for a lag of 90 degrees.

degrees remain undetected. For lags larger than 45 degrees, its sensitivity starts to increase and is maximal at a lag of 90 degrees. $T_{\text{real}}$ essentially behaves in the opposite way: it is highest at lag zero and decreases towards zero with increasing lag. In particular, interactions with lags larger than about 80 degrees remain undetected. Important to note is that the non-zero values of $T_{\text{real}}$ for lags between 45 and 80 degrees are in fact spurious and when these are corrected for, $T_{\text{real}}$ essentially behaves the same as $T_{\text{imag}}$ but reflected in the 45 degrees axis.

The properties of $T_{\text{imag}}$ and $T_{\text{real}}$ considered above are *local* in the sense that we only considered their behavior at a single pair of system locations. Although locally they behave the same, up to reflection, *globally* they behave very differently and this is in fact the reason why imaginary test-statistics have become popular in neuroscience [11–15]. We first consider the global behavior of the imaginary test-statistic. We consider the extreme cases of $\phi = 0$ degrees and $\phi = 90$ degrees. The sensitivities for all source-pairs are shown in Fig 2C (0 degrees) and Fig 2E (90 degrees). In the case of $\phi = 0$ degrees, none of the source-pairs show significant interaction. Thus, on the one hand, the true interaction at $(k_0, l_0)$ remains undetected, but on the other hand, no spurious interactions are present at any other source-pair. In the case of $\phi = 90$ degrees, sensitivity is highest at the location $(k_0, l_0)$ of true interaction and is otherwise low. Thus, also in this case, spurious interactions are (almost) absent. The global behavior of $T_{\text{imag}}$ therefore allows for the detection of interactions as long as the lag is not too small. Fig 2D

shows that the global behavior of $T_{\text{real}}$ is very different. In particular, for both $\phi = 0$ and $\phi = 90$ degrees, their are many source pairs that exhibit spurious interactions, especially around the true source locations. Moreover, although for $\phi = 0$ degrees, the test-statistic is weakly sensitive to the true interaction, in practice it will not be detected because of the presence of spurious interactions. Thus, $T_{\text{real}}$ does not allow for the detection of interactions, irrespective of the lag. Fig 2B also shows that, locally, the complex-valued test-statistic $T_{\text{compl}} = \hat{\gamma}_{k_0, l_0}$ is roughly uniformly sensitive to interactions i.e. it is non-zero and roughly equal for all lags. Unfortunately, the global behavior of $T_{\text{compl}}$ does not allow for the detection of true interactions because of the high level of spurious connectivity in its real part.

## Subspace projections for suppressing spurious interactions

The goal of this study is to reduce the spurious connectivity in $T_{\text{compl}}$ so that it allows for the detection of system interactions at all lags. Spurious connectivity will be reduced by correcting the test-statistics using appropriately defined subspace projections. We consider three types of corrections and the associated corrected source-space spectral matrices. The matrices differ in the order in which the inverse operator and the projection operators are applied and in which space the subspace projection is applied (source- or sensor-space). The first type is obtained by first projecting the time-frequency coefficients of the sensor data to source-space by using an arbitrary linear inverse operator. Subsequently, the reconstructed time-frequency coefficients of the source-activity are used to estimated the activity's spectral matrix. This is done using the sample spectral matrix since this is an unbiased estimator. Lastly, the estimated spectral matrix is corrected by applying a suitably defined subspace projection. Note that this projection is defined in the vectorspace of source-space spectral matrices. Fig 3A provides an illustration. The second type is obtained by switching the order of the inverse and subspace projection operators as illustrated in Fig 3B. The second type is also obtained by switching the order, but applying the subspace projection in sensor-space instead of source-space as illustrated in Fig 3C. The third of the three types of corrections described above has been proposed in [27]. Note that whereas the source-based estimator makes use of the reconstructed source-activity, the two sensor-based estimators avoid this step by applying an inverse operator in the space of spectral matrices instead of in signal space. The difference between the two sensor-based estimators is in which space the subspace projection is applied (source- or sensor-space). We will provide a comparative performance analysis of these three types of corrected estimators.

## Estimation of system interactions

### Source-space based estimation

Measurements in the time-frequency domain at a particular frequency $\omega$ yield a $n \times k$ complex-valued data matrix $Y$, where $n$ and $k$ denote the number of sensors and samples, respectively. The observed data is related to source activity $X$ and sensor noise $E$ through

$$Y = LX + E,$$

where $X$ and $E$ are complex-valued $p \times k$ and $n \times k$ matrices, respectively. If $X$ could be observed directly, the best estimator for the spectral matrix $S_x$ is the sample spectral matrix $\hat{S}_x = XX^{\dagger}/(n-1)$. When only $Y$ can be observed directly, a natural way to estimate $S_x$ is to first reconstruct the source activity $X$ be applying an inverse operator to the observed sensor data and subsequently to compute the sample spectral matrix of the reconstructed source activity. Let $X^{\star} = L^{\sharp} Y$ be the reconstructed source activity, where $L^{\sharp}$ is some linear inverse

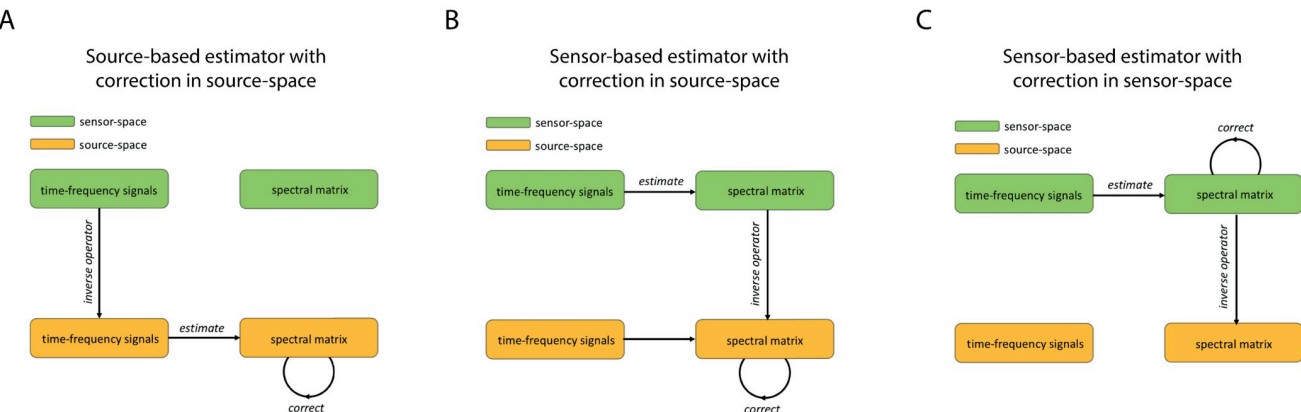

**Fig 3. Subspace projections for suppression of spurious interactions.** A. Source-based estimator with correction in source-space. First the time-frequency coefficients of the sensor data are projected to source-space using an arbitrary linear inverse operator. Next the spectral matrix of the reconstructed time-frequency coefficients are estimated and subsequently corrected by applying a subspace projection. B. Sensor-based estimator with correction in source-space. First the spectral matrix of the sensor-space time-frequency coefficients is estimated. Next, the estimated sensor-space spectral matrix is projected to source-space using the tensor product of an arbitrary linear inverse operator with itself and subsequently corrected by applying a subspace projection. C. Sensor-based estimator with correction in sensor-space. First the spectral matrix of the sensor-space time-frequency coefficients is estimated. Next, the estimated sensor-space spectral matrix is corrected by applying a subspace projection and subsequently projected to source-space using the tensor product of an arbitrary linear inverse operator with itself.

operator, and define the estimator

$$\hat{S}_1 = \frac{1}{n-1} X^\star (X^\star)^\dagger = L^\sharp \hat{S}_y (L^\sharp)^\dagger,$$

where $\hat{S}_y = YY^\dagger/(n-1)$ denotes the sample spectral matrix of the observed sensor activity. Note that $\hat{S}_1$ is Hermitian as required for a proper estimator of $S_x$. We will refer to this estimator as *source-based* to emphasize that it is defined in terms of the (reconstructed) source activity. It will be convenient to write it in vectorized form as

$$Vec(\hat{S}_1) = (L^\sharp \otimes L^\sharp) Vec(\hat{S}_y),$$

where $\otimes$ denotes the (Kronecker) tensor product. The estimator can be expressed in terms of $X$ and $E$ by first writing

$$Vec(\hat{S}_y) = (L \otimes L) Vec(\hat{S}_x) + Vec(\hat{S}_e) + 2(I_n \otimes L) Vec(\text{Re}(\hat{S}_{x,e})), \quad (3)$$

where $\hat{S}_e = EE^\dagger/(n-1)$ denotes the sample spectral matrix of the measurement noise and $\hat{S}_{x,e} = XE^\dagger/(n-1)$ denotes the sample cross-spectral matrix between source activity and measurement noise. Consequently,

$$Vec(\hat{S}_1) = R_1 Vec(\hat{S}_x) + F_1 Vec(\hat{S}_e) + 2F_1(I_n \otimes L) Vec(\text{Re}(\hat{S}_{x,e})),$$

where $R_1 = (L^\sharp L \otimes L^\sharp L)$, and $F_1 = L^\sharp \otimes L^\sharp$ are the associated resolution and inverse operators, respectively. Since source activity and measurement noise are uncorrelated and $\hat{S}_x$ and $\hat{S}_e$ are unbiased estimators of $S_x$ and $S_e$, respectively, the expectation $\langle \hat{S}_1 \rangle$ of $\hat{S}_1$ is

$$Vec(\langle \hat{S}_1 \rangle) = R_1 Vec(S_x) + F_1 Vec(S_e).$$

We now compute the filtering properties of $\hat{S}_1$, which will be used in the next section to establish a relationship between the source- and sensor-based estimators. Let $r \leq n$ be the rank

of $L$ and let $L = UDV^T$ be the singular value decomposition of $L$, where $U$ is an $n \times r$ orthonormal matrix, $V$ is an $p \times r$ orthonormal matrix, and $D$ is the $r \times r$ diagonal matrix that holds the non-zero singular values $\delta_1 \geq \delta_2 \geq \cdots \geq \delta_r$ of $L$ in decreasing order. Note that $U$ and $V$ are bases for sensor and source space, respectively. Using the singular basis of the forward matrix, the resolution operator $R_1$ can be represented as

$$R_1(\lambda_1) = \sum_{k=1}^{r} \sum_{l=0}^{r} f_{k,l}^{(1)}(\lambda_1)(v_k u_l^T) \otimes (v_k u_l^T),$$

where the *filter coefficients* $f_{k,l}^{(1)}(\lambda_1)$ are given by

$$f_{k,l}^{(1)}(\lambda_1) = \frac{\delta_k^2 \delta_l^2}{(\delta_k^2 + \lambda_1)(\delta_l^2 + \lambda_1)}.$$

The filter coefficients range between zero and one and measure to what extent a contribution of the form $v_k u_l^T$ of the true source spectral matrix is retained in the reconstructed spectral matrix. In particular, the diagonal entries of the filter matrix (i.e. the gains) are

$$f_{jj}^{(1)}(\lambda) = \frac{\delta_j^2}{(\delta_j^2 + \lambda)^2}.$$

## Sensor-space based estimation

The source-space based estimator is obtained by first projecting the observed sensor data to source-space and subsequently computing the sample spectral matrix of the reconstructed source-activity. An alternative way of estimating the source-space spectral matrix is to first estimate the sensor-space spectral matrix and subsequently to project it to source-space [27]. We will refer to this estimator as *sensor-based* and denote it by $\hat{S}_2$. It is obtained by writing the sample sensor-space spectral matrix in vectorized form and subsequently projecting it to source-space by using the inverse operator $F_2 = (L \otimes L)^\sharp$:

$$Vec(\hat{S}_2) = F_2 Vec(\hat{S}_y),$$

where

$$F_2 = (L \otimes L)^\sharp = (L \otimes L)^T [(L \otimes L)(L \otimes L)^T + \lambda I]^{-1}.$$

Alternatively, this estimator can also be derived by solving the following convex optimization problem:

$$\arg \min \; \|\hat{S}_y - L S_x L^T\|^2 + \lambda_2 \|\hat{S}_y\|^2,$$

where the minimization is done over $S_x$. The gradient of this objective function has a unique minimum in $\hat{S}_2$.

The sensor-based estimator can be written in terms of $X$ and $E$ as

$$Vec(\hat{S}_2) = R_2 Vec(\hat{S}_x) + F_2 Vec(\hat{S}_e) + 2F_2(I_n \otimes L) Vec(\text{Re}(\hat{S}_{x,e})),$$

where $R_2 = F_2(L \otimes L)$ is the associated resolution operator. Furthermore, its expectation it given by

$$Vec(\langle \hat{S}_2 \rangle) = R_2 Vec(S_x) + F_2 Vec(S_e).$$

Comparing the expressions for $\hat{S}_1$ and $\hat{S}_2$, we see that both are obtained by applying a linear inverse operator to the vectorized sensor-data $Vec(\hat{S}_y)$: For the source-based estimator, the inverse operator is $L^\sharp \otimes L^\sharp$ and for the sensor-based estimator, the inverse operator is $(L \otimes L)^\sharp$. It is easily verified that in the absence of regularization (i.e. $\lambda = 0$), both estimators reduce to the Moore-Penrose inverse of $L \otimes L$. In the limit of strong regularization, the inverse operators corresponding $\hat{S}_1$ and $\hat{S}_2$ reduce to $\lambda^{-1}(L \otimes L)$ and $\lambda^{-2}(L \otimes L)$, respectively.

The general relation between $\hat{S}_1$ and $\hat{S}_2$ is obtained by comparing their filtering coefficients. The resolution operator $R_2$ corresponding to $\hat{S}_2$ can be represented as

$$R_2(\lambda_2) = \sum_{k=1}^{r}\sum_{l=0}^{r} f_{k,l}^{(2)}(\lambda_2)(v_k u_l^T) \otimes (v_k u_l^T),$$

with filter coefficients

$$f_{k,l}^{(2)}(\lambda_2) = \frac{\delta_k^2 \delta_l^2}{\delta_k^2 \delta_l^2 + \lambda_2}.$$

In the absence of regularization, the filter coefficients of both estimators are equal to one, so that all contributions of the form $v_k u_l^T$ are completely retained. The gains are

$$f_{jj}^{(2)}(\lambda) = \frac{\delta_j^2}{\delta_j^4 + \lambda^2}.$$

Because of the extra term $2\lambda\delta_j^2$ in the numerator of $f_{jj}^{(1)}$, the source-based estimator provides stronger damping than the sensor-based estimator. The expressions for the filter coefficients also show that there is a natural correspondence between $\lambda_1$ and $\lambda_2^2$, which allows the estimators to be directly compared by setting $\lambda_2 = \lambda_1^2 = \lambda^2$.

## The geometry of signal leakage

*Signal leakage* refers to bias in reconstructions of brain activity that is due to the incomplete unmixing of sensor signals by application of an inverse operator. When measuring electric potentials as in EEG, ECoG, and LFP, signal leakage is called *volume conduction* and when measuring magnetic fluxes such as in MEG, signal leakage is called *field spread* [6]. Here we provide a geometric characterization of signal leakage that pertains to the functional connectivity estimators defined in *textbfEstimation of system interactions* and that applies to any linear inverse operator $L^\sharp$. This characterization will form the basis for defining functional connectivity estimators that actively suppress signal leakage.

The expectations of the estimators defined in ***Estimation of system interactions*** are of the form

$$Vec(\langle \hat{S}_j \rangle) = R_j Vec(S_x) + F_j Vec(S_e), \tag{4}$$

for $j = 1, 2$. The second term on the right-hand-side of Eq (4) corresponds to signal leakage due to measurement noise that is projected to source-space by the inverse operator. The first term on the right-hand-side contains true source-space functional connectivity in vectorized form, but in a distorted way due to multiplication by $R \otimes R$. For a square matrix $M$, let $M^+$ and $M^-$ denote the matrices obtained from $M$ by setting its off- and on-diagonal entries, respectively, to zero. Thus, $M$ can be decomposed as $M = M^+ + M^-$ and the same is true for its

vectorization. We can hence write the expectations as

$$Vec(\langle \hat{S}_j \rangle) = R_j Vec(S_x^+) + R_j Vec(S_x^-) + F_j Vec(S_e). \tag{5}$$

We now interpret each of the three terms on the right-hand-side of Eq (5).

Note that the first term on the right-hand-side of Eq (5) only depends on the diagonal entries of $S_x$, i.e. on the power of (active) sources and not on their interactions. In this sense, all non-zero off-diagonal entries correspond to spurious connectivity. We refer to this term as the *leakage term*. Eq (5) shows that the leakage term lies in the subspace of $\mathbb{C}^{p^2}$ that is spanned by the $1 + (i - 1)(p + 1)$-th columns of $R_j$ for $1 \leq i \leq p$. We write $B_j$ for the matrix that contains these columns of $R_j$ and refer to the column space of $B_j$ as the *leakage subspace*. The leakage term can be written explicitly as

$$R_j Vec(S_x^+) = \sum_{i=1}^{p} \sigma_i^2 r_i^{(j)} \otimes r_i^{(j)},$$

where $\sigma_i^2$ denotes the power of the $i$-th source and $r_i^{(j)}$ denotes the $i$-th point-spread function of $R_j$.

The second term on the right-hand-side of Eq (5) does not depend on source power but only on the interactions between (active) sources. Its off-diagonal entries are non-spurious in the sense that they vanish in the absence of interactions. Its entries are generally biased, however, except when $r_i = e_i$ for $1 \leq i \leq p$, which is impossible since $n < p$. We refer to this term as the *interaction term*. Eq (5) shows that it lies in the subspace of $\mathbb{C}^{p^2}$ that is spanned by the columns of $R_j$ that are not in $B_j$. We refer to this space as the *connectivity subspace*. The interaction term can be written explicitly as:

$$R_j Vec(S_x^-) = \sum_{i < i'} \gamma_{ii'}(r_i^{(j)} \otimes r_{i'}^{(j)} + r_{i'}^{(j)} \otimes r_i^{(j)}),$$

where $\gamma_{ii'}$ denotes the $(i, i')$-th entry of $S_x$ i.e. the interaction between source $i$ and $i'$, and where the sum runs over all ordered pairs $(i, i')$ with $i < i'$.

Lastly, the third term on the right-hand-side of Eq (5) is independent of source activity and only depends on the measurement noise and as such is entirely spurious. We refer to this term as the *(projected) noise term*. Eq (5) shows that it is contained in the column space of $F_j$, which we will refer to as the *(projected) noise subspace*. It can be written explicitly as

$$F_j Vec(S_e) = \sum_{i \leq i'} \mu_{ii'} f_i^{(j)} \otimes f_{i'}^{(j)},$$

where $\mu_{ii'}$ denotes the $(i, i')$-th entry of $S_e$ and $f_i^{(j)}$ denotes the $i$-column of $F_j$, which can be thought of as a noise point-spread function of the $i$-th sensor. The sum runs over all ordered pairs $(i, i')$ with $i \leq i'$.

To illustrate the effects of signal leakage on the estimated spectral matrix $\hat{S}_x$, we consider two simple source configurations in the absence of measurement noise. We first consider a pair of point sources at $k$ and $l$, with variances $\sigma_k^2$ and $\sigma_l^2$, respectively, that interact with strength $\gamma$. Thus, the leakage and interaction terms are given by

$$\sigma_k^2 r_k \otimes r_k + \sigma_l^2 r_l \otimes r_l^T,$$

and

$$\gamma(r_k \otimes r_l + r_k \otimes r_l),$$

respectively. The squared norm of the estimated leakage term therefore is

$$\|\sigma_k^2 r_k \otimes r_k + \sigma_l^2 r_l \otimes r_l^T\|^2 = \sigma_i^4 \|r_i\|^4 + \sigma_j^4 \|r_j\|^4 + 2\sigma_i^2 \sigma_j^2 \langle r_i, r_j \rangle^2,$$

where the brackets $\langle , \rangle$ denote the standard inner product, and the squared norm of the estimated connectivity term is given by

$$\|\gamma(r_k \otimes r_l + r_k \otimes r_l)\|^2 = 2\gamma^2(\langle r_i, r_j \rangle^2 + \|r_i\|^2 \|r_j\|^2).$$

By comparing these expressions and using the fact that $\sigma_1^2 \sigma_2^2 \geq \gamma^2$, which follows from the positive semi-definiteness of $S_x$, we see that the inequality

$$\|\sigma_k^2 r_k \otimes r_k + \sigma_l^2 r_l \otimes r_l^T\| \geq \|\gamma(r_k \otimes r_l + r_k \otimes r_l)\|,$$

is implied by

$$(\sigma_k^2 \|r_k\|^2 - \sigma_l^2 \|r_l\|^2)^2 \geq 0,$$

which is trivially true. Therefore, for two interacting point sources, signal leakage is always stronger than (estimated) interaction strength.

As a second example we consider a spatially extended source covering $m$ locations. If the source is not too large, then $r_k \approx r_l$ for all locations $k$, $l$ within the source. Furthermore, if the source is homogeneous, say with variance $\sigma^2$, then the leakage term is approximately equal to $m\sigma^2 \, r \otimes r$, where $r$ denotes the shared point-spread function within the source ($r \approx r_k \approx r_l$). Moreover, if the interactions within the source are homogeneous, say with shared strength $\gamma$, then the connectivity term is approximately equal to $\gamma m(m-1) r \otimes r$. Thus, the strength of signal leakage, relative to that of the estimated connectivity, is approximately equal to $\sigma^2/\sqrt{(m-1)\gamma^2}$. This shows that for spatially extended sources (i.e. $m$ large), the estimated intra-source connectivity is dominated by interaction effects and signal leakage can hence be neglected. In particular, suppressing signal leakage is not necessary in this situation.

## Suppression of signal leakage

### Suppression in source-space

In **_The geometry of signal leakage_** we showed that $Vec(\langle \hat{S}_j \rangle)$ can be decomposed into three terms that lie in different linear subspaces of $\mathbb{C}^{p^2}$ and correspond to leakage, interaction, and projected noise, respectively. We now define a sequence of linear operators acting on $\mathbb{C}^{p^2}$ that increasingly suppress the leakage term, while retaining the interaction term as much as possible. To construct this sequence, we exploit the fact that the leakage subspace is spanned by the columns of the matrix $B_j$. Thus, if $B_j = U_j \Lambda_j V_j^T$ is a singular value decomposition of $B_j$ and the dimension of the leakage subspace is $d_j$, the first $d_j$ columns of $U_j$ form an orthonormal basis for the leakage subspace. For $0 \leq k \leq d_j$, we define the linear operator $\pi_j^{(k)}$ on $\mathbb{C}^{p^2}$ by

$$\pi_j^{(k)} = I - U_j^{(k)}(U_j^{(k)})^T,$$

where $U_j^{(k)}$ is the matrix carrying the first $k$ columns of $U_j$. Note that $\pi_j^{(k)}$ is the projection onto the orthogonal complement of the span of the first $k$ left-singular basis vectors of $B_j$. In the case $k = 0$, the projection reduces to the identity operator and in the case $k = d_j$, the projection nullifies any vector that lies in the leakage subspace.

We now define a sequence of corrected interaction estimators by

$$Vec(\hat{S}_j^{(k)}) = \pi_k Vec(\hat{S}_j),$$

for $0 \leq k \leq d_j$. Note that $\hat{S}_j^{(0)} = \hat{S}_j$ and that larger projection ranks $k$ provide increasingly stronger suppression of the leakage term. In particular, for $k = d_j$, the leakage term is completely suppressed. The expected value of the corrected estimator is given by

$$Vec(\langle \hat{S}_j^{(k)} \rangle) = R_j^{(k)} Vec(S_x) + F_j^{(k)} Vec(S_e),$$

where $R_j^{(k)} = \pi_j^{(k)} R_j$ is the corrected resolution operator and $F_j^{(k)} = \pi_j^{(k)} F_j$ is the corrected inverse operator.

An interesting property of the operator $\pi_j^{(k)}$ is that it does not alter the imaginary part $R_j \mathrm{Im}(S_x^-) R_j^T$ of the interaction term $R_j S_x^- R_j^T$. This follows from the fact that the subspaces of symmetric and anti-symmetric matrices are orthogonal to each other. Indeed, the column space of $U_j^{(k)}$ is contained in the space of symmetric matrices, so the space onto which $\pi_j^{(k)}$ projects contains the space of anti-symmetric matrices. Since the imaginary part of the interaction term is anti-symmetric, it is invariant under the action of $\pi_j^{(k)}$. Thus, $\pi_j^{(k)}$ does not suppress the imaginary, i.e. lagged part of the system interactions. In the rest of the text, we will refer to this type of suppression as *suppression in source-space*. Also note that a global scaling of the interaction-strength scales the projected interaction term by the same factor and leaves the projected leakage term unchanged. Specifically, let $\mu \geq 0$ be a scaling factor and consider the scaled source-space spectral matrix $S_x = S_x^+ + \mu S_x^-$, then

$$Vec(\langle \hat{S}_j^{(k)} \rangle) = R_j^{(k)} Vec(S_x^+) + \mu R_j^{(k)} Vec(S_x^-) + F_j^{(k)} Vec(S_e).$$

## Suppression in sensor-space

As proposed in [27], a similar correction can be applied in sensor-space. Instead of first projecting the sensor-data to source-space and subsequently computing and correcting the sample source-space spectral matrix, first the sample spectral matrix of the observed sensor-data is computed and corrected, and subsequently projected to source-space. From Eq (3) it follows that the expected value of the vectorized sample sensor-space spectral matrix is given by

$$Vec(\langle \hat{S}_y \rangle) = (L \otimes L) Vec(S_x) + Vec(S_e),$$

which can be decomposed as

$$Vec(\langle \hat{S}_y \rangle) = (L \otimes L) Vec(S_x^+) + (L \otimes L) Vec(S_x^-) + Vec(S_e),$$

similar to the decomposition of $Vec(\langle \hat{S}_j \rangle)$ in Eq (5). This allows to define sensor-level equivalents of the leakage, interaction, and projected noise subspaces defined in ***The geometry of signal leakage*** as well as a sensor-level equivalent of the leakage correction $\pi_j^{(k)}$. Concretely, let $B$ be the matrix carrying the $1 + (i-1)(p+1)$-th columns of $L \otimes L$ for $1 \leq i \leq p$, let $B = U \Lambda V^T$ be a singular value decomposition of $B$, and let $d_3$ denote the dimension of the (sensor-level) leakage subspace. Then the first $d_3$ columns of $U$ form an orthonormal basis for the (sensor-level) leakage subspace and we hence can define the projection $\pi_3^{(k)}$ on $\mathbb{C}^{n^2}$ by

$$\pi_3^{(k)} = I - U^{(k)} (U^{(k)})^T,$$

where $U^{(k)}$ is the matrix carrying the first $k$ columns of $U$. Completely analogous to the

definition in **Suppression in source space**, we define a sequence of corrected sensor-space spectral estimators by

$$Vec(\hat{S}_y^{(k)}) = \pi_3^{(k)} Vec(\hat{S}_y),$$

for $1 \leq k \leq d_3$, and subsequently project these to source-space to obtain the following corrected estimator for the source-space spectral matrix:

$$Vec(\hat{S}_3^{(k)}) = F^{(k)} Vec(\hat{S}_y^{(k)}),$$

where the inverse operator $F_3^{(k)}$ is given by

$$F_3^{(k)} = (L \otimes L)^T \pi_3^{(k)} [\pi_3^{(k)} (L \otimes L)(L \otimes L)^T \pi_3^{(k)} + \lambda I]^{-1} \pi_3^{(k)}.$$

Note that $\hat{S}_3^{(0)} = \hat{S}_2$ so that $\hat{S}_2^{(k)}$ and $\hat{S}_3^{(k)}$ correspond to different ways to correct $\hat{S}_2$. The expected value of $\hat{S}_3^{(k)}$ is given by

$$Vec(\langle \hat{S}_3^{(k)} \rangle) = R_3^{(k)} Vec(S_x) + F_3^{(k)} Vec(S_e),$$

where $R_3^{(k)} = F_3^{(k)}(L \otimes L)$ is the associated resolution operator. Thus, the expected value of the sensor-space corrected estimator has the same form as the source-space corrected estimators and therefore has the same leakage structure in source-space. We do observe that, in contrast to the source-space corrected estimators, the norm of $\langle \hat{S}_3^{(k)} \rangle$ does not need to be a decreasing function of $k$. In particular, although for $k = d_3$ the sensor-space leakage term $(L \otimes L) Vec(S_x^+)$ is completely suppressed, this need not be true for the source-space leakage term $R_3^{(k)} Vec(S_x^+)$. Like the sensor-space corrected estimators, the imaginary part of the system interactions is not suppressed. In the rest of the text, we will refer to this type of suppression as *suppression in sensor-space*.

## Effectiveness of suppression

The effectiveness of the projections defined in the previous two sections in suppressing signal leakage in source-space interactions does not only depend on the extent to which the leakage term is suppressed, but also on the extent to which the interaction term is suppressed. Indeed, because the leakage, interaction, and noise terms are not separately observable, the projections need to be applied to their sum, i.e. the estimated spectral matrix. Consequently, they might not only suppress leakage, but also partly suppress interactions and projected sensor noise. The strengths of the corrected leakage and interaction terms, relative to those of the respective uncorrected terms can be quantified using the generalized Rayleigh quotient. Let $S$ be any source-space spectral matrix. We consider the generalized Rayleigh quotient $Q_j^{(k)}$ with positive semi-definite matrices $G_j^{(k)}$ and $G_j$ where

$$G_j^{(k)} = (R_j^{(k)})^T R_j^{(k)}$$

is the Gram matrix of the corrected resolution operator $R_j^{(k)}$. Thus, for any source-space spectral matrix $S$, the generalized Rayleigh quotient is given by

$$Q_j^{(k)}(S) = \frac{Vec(S)^\dagger G_j^{(k)} Vec(S)}{Vec(S)^\dagger G_j Vec(S)} = \frac{\|R_j^{(k)} Vec(S)\|^2}{\|R_j Vec(S)\|^2}.$$

One minus the Rayleigh quotient quantifies the extent to which the expected value $\langle \hat{S}_j^{(k)} \rangle$ of the uncorrected estimator is reduced by the correction $\pi_k^{(k)}$. Below we will refer to this as the *suppression level*. The suppression levels of the leakage and interaction terms are defined as $1 - Q_j^{(k)}(S^+)$ and $1 - Q_j^{(k)}(S^-)$, respectively. The fact that $\pi_j^{(k)}$ does not suppress the imaginary part of the system interactions can be expressed as $Q_j^{(k)}(\mathrm{Im}(S)) = 1$ for all $k$. For the source-space corrections (i.e. $j = 1, 2$) the range of $Q_j^{(k)}$ is confined to the unit interval because $G_j^{(k)}$ then reduces to $R_j^T \pi_j^{(k)} R_j$. Furthermore, complete suppression of $S$ by a rank-$k$ correction is equivalent to $\langle \hat{S}_j \rangle$ being contained in the column space of $U_j^{(k)}$ and no suppression is equivalent to $\langle \hat{S}_j \rangle$ being contained in the orthogonal complement of $U_j^{(k)}$. In the ideal case, the leakage term is completely suppressed and the interaction term is entirely retained, i.e. $Q_j^{(k)}(S^+) = 0$ and $Q_j^{(k)}(S^-) = 1$ for some $k$. This can also be interpreted geometrically by noting that the angle $\alpha_k(S)$ between $R_j \, Vec(S)$ and the column space of $U_k$ equals

$$\alpha_k(S) = \sin^{-1}\sqrt{Q_j^{(k)}(S)},$$

which shows that the suppression level and $\alpha_k(S)$ are related through a monotonically decreasing function.

## Comparative performance

### Effectiveness

We calculated the suppression levels of the interaction term as a function of the regularization-level $\lambda$ for five increasing interaction lags of 0, 20, 40, 50, and 70 degrees. Suppression of interactions with a lag of 90 degrees is zero because the imaginary part of interactions is invariant under the action of the correction projection. All other parameter values were set as in Table 1. The projection ranks were set to their maximal values so that the leakage terms are completely suppressed. Fig 4A shows the results for the source-based estimator. Observe the presence of three regimes: An under-regularization regime ($\lambda < -2$) in which the interaction is not suppressed, a regime in which the level of regularization is appropriate ($-2 < \lambda < 1$) and in which the level of suppression sensitively depends on the value of $\lambda$, and an over-regularization regime ($\lambda > 1$) in which the interaction is completely suppressed. This behavior can be observed for all lags and for the sensor-based estimators as well (see Fig 4B and 4C). The absence of suppression in the under-regularization regime implies that the interaction term is contained in the orthogonal complement of the leakage subspace. As $\lambda$ increases, the angle between the interaction term and the leakage subspace decreases and this is reflected in stronger suppression. For large values of $\lambda$, the angle approaches zero so that the interaction term will be contained in the leakage subspace and hence will be completely suppressed. These results make clear that the effectiveness of the correction decreases with increased regularization-levels. In practice this means that the correction is less effectiveness for noisy data. It is therefore not advised to use regularization-levels that have been optimized for the estimation of source power instead of source interactions. For [35] have demonstrated that the optimal regularization-level for interactions is lower than that for source-power, at least for the ridge inverse operator (see also [36]).

Fig 4 also shows that for all three estimators and for all regularization-levels, suppression levels decrease with increasing interaction lag $\phi$. This is a consequence of the imaginary part of the interactions being unaffected by the corrections and can be understood in the following way. Let $s = \gamma\cos\phi + i\gamma\sin\phi$ be the interaction with strength $\gamma$ and lag $\phi$. The effect of the

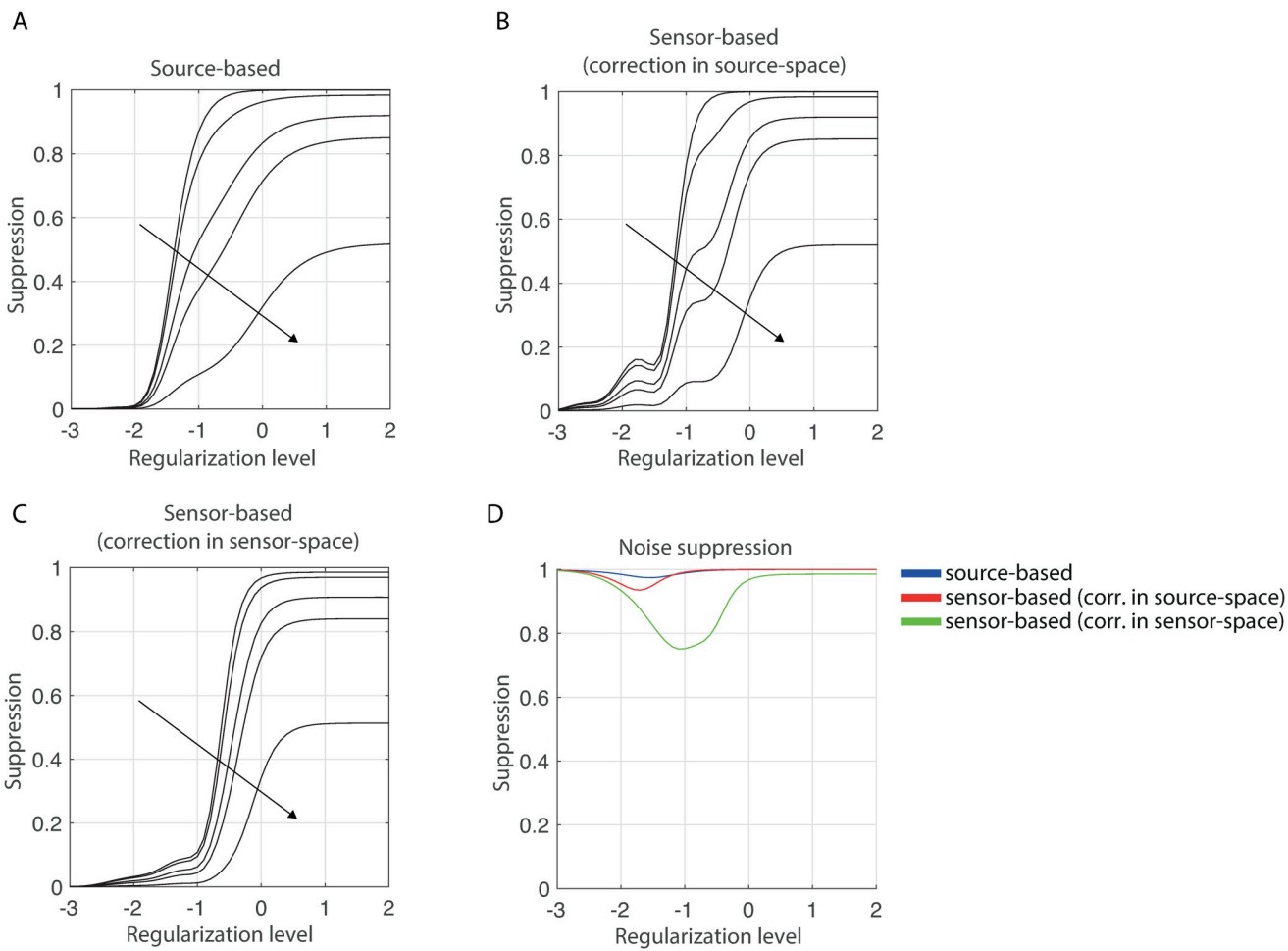

**Fig 4. Suppression levels.** A. Suppression levels of the source-based estimator as a function of regularization-level and for three interaction lags (0, 20, 40, 50, and 70 degrees). B. Same as in A. but for the sensor-based estimator with suppression in source-space. C. Same as in A. but for the sensor-based estimator with suppression in sensor-space. D. Suppression of projected sensor noise for all three estimators and as a function of regularization-level. In panels A, B, and C, the interaction lags increase in the direction of the arrows. In all panels, the parameters were set as in Table 1.

correction on $s$ is that its real part is suppressed, and its imaginary part is retained. Hence, applying the correction to $s$ gives $r\gamma\cos\phi + i\gamma\sin\phi$, for certain $0 \leq r \leq 1$, where $r = 1$ and $r = 0$ correspond to no and complete suppression, respectively. The suppression level is now given by

$$\frac{1 - r^2}{1 + \tan^2\phi},$$

which is a decreasing function of the interaction lag $\phi$. Since the suppression levels are (roughly) increasing functions of the regularization-level, not exceeding a given upper-limit on suppression requires weaker regularization-levels for smaller lags. For example, for the source-based estimator (see Fig 4A) not exceeding a suppression level of 0.3 requires a regularization-level of at most $\lambda = 0$, if the lag is 70 degrees. However, for instantaneous interactions, it requires a regularization-level of at most $\lambda = -1.5$. In practice this means that the detection of interactions with small lags requires data that is less noisy.

Fig 4D shows the suppression levels of the projected sensor noise for all three estimators. The projected noise term is independent of system activity and its suppression level is therefore independent of the lag. Observe that for weak and strong regularization-levels the noise is completely suppressed, implying that for these levels, the projected noise term is contained in the leakage subspace. For intermediate regularization-levels, the noise term is somewhat less suppressed, particularly in the case of the sensor-based estimator with correction applied in sensor-space. For the other two estimators, the noise is always suppressed stronger than the interaction term is, hence correction in source-space increases the signal-to-noise ratio of the estimated source-space spectral matrix.

## Bias reduction

Here we assess to what extent the biases of the estimators are reduced by the leakage corrections. We here understand "bias" to be any function of the to-be-estimated parameter and the expected value of the estimator. Different functions capture different aspects of the estimators' expected value, some of which might be more relevant more a particular application than others. We quantified the bias by the ratio of the estimators' absolute expected value, averaged over all but the true interaction-pair, to itself plus the absolute expected value at the true interaction-pair $(k_0, l_0)$. In calculating the numerator, pairs of the form $(j, j)$ were excluded since these pertain to source power and not to source interaction. Furthermore, to take into account the limited resolution of the inverse operators, we relaxed the definition of "true interaction-pair" to include the eight surrounding pairs $(k_0 \pm 1, l_0 \pm 1)$. The bias takes on values between zero and one, where zero corresponds to the estimators' expected value being zero at all but the true interaction-pair and one corresponds to the estimators' expected value being zero at the true interaction-pair. We computed the bias as a function of noise-level. Below, we refer to "performance" as one minus the bias. As in the previous section, the projection ranks of the corrected estimators were set to their maximal values so that signal leakage is completely suppressed. To isolate the effect of the correction from the issue of selecting an appropriate regularization-level, we let the regularization-level range between −8 and 0 in steps of 0.5 and selected the value that maximized performance. This was done for both the uncorrected and corrected estimators. The interaction lag was set to $\phi = 0$ degrees and all other parameter values were chosen as in Table 1.

Fig 5A (blue and red bars, respectively) shows the performance of the uncorrected source- and sensor-based estimators as a function of noise-level. Performance decreases with increasing noise-level, which is to be expected since the norm of the projected sensor noise is proportional to the noise-level. Also notice that the performance of the sensor-based estimator decreases slower with increasing noise-level than that of the source-based estimator. The reason for this lies in the different filtering properties of the respective inverse operators that are amplified for increasing regularization levels. These differences cause the sensor-based estimator to have a somewhat higher spatial resolution than the source-based estimator. This is illustrated in Fig 5B and 5C, which show the expected values of the source- and sensor-based estimators, respectively, for $\sigma = 0.05$ and $\lambda = 10^{-2.2}$. The true interaction-pair is denoted by the white circles. Note that the interaction is a bit better visible in the sensor-based estimator, which has a somewhat higher resolution. In particular, the reconstructed sources and their interactions are a bit more localized than in case of the source-based estimator. On the other hand, the sensor-based reconstruction seems to suffer a bit more from artifacts than the source-based estimator, which could affect statistical significance testing. Lastly, we observe that the biases of both estimators are relatively large. For example, a bias of 0.5 means that the average amplitude of the reconstruction at the true source-pair and that at other source-pairs

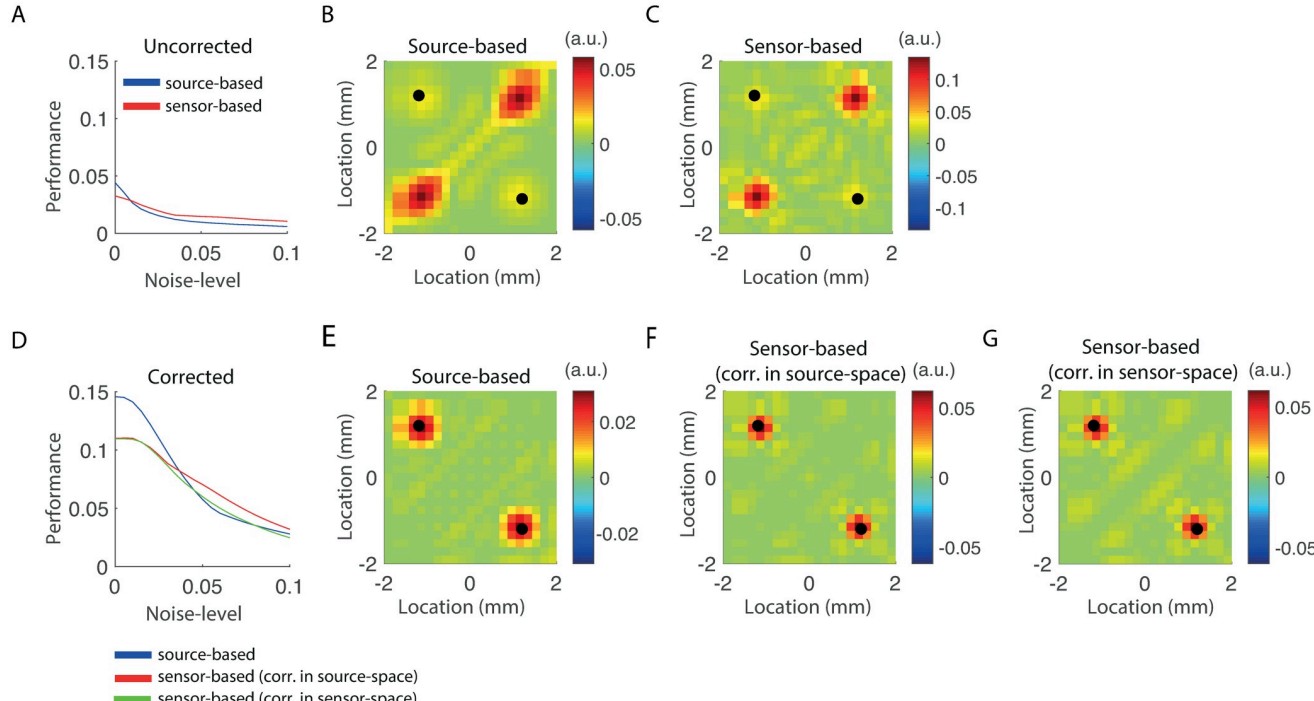

**Fig 5. Bias-reduction through leakage correction.** A. Performance on the test-model of the uncorrected estimators as a function of noise-level. Performance is defined as one minus the bias and ranges between zero and one. B. Expected value of the source-based estimator for $\sigma = 0.05$ and $\lambda = -2.2$. C. Same as B but for the sensor-based estimator. D. Performance on the test-model of the corrected estimators as a function of noise-level. E. Expected value of the corrected source-based estimator for $\sigma = 0.05$ and $\lambda = -3$. F. Same as E but for the sensor-based estimator with correction in source-space. G. Same as E but for the sensor-based estimator with correction in sensor-space. In panels B, C, E, F, and G, the true interaction-pair is designated by the white-circles.

are equal. In practice this means that the interaction will not be detected by pair-wise tests. The main reason for the high biases is the presence of signal leakage around the seed locations. The performance of the corrected estimators are shown in Fig 5D. Observe that the performance of all three corrected estimators is several times higher than that of both uncorrected estimators. The differences are largest in the absence of noise, but can present across noise-levels. To illustrate the differences, we calculated the expectation values of the corrected estimators for $\sigma = 0.05$ and $\lambda = 10^{-3}$. They are shown in Fig 5 (panels E, F, and G). The interaction stands out clearly, due to the complete suppression of signal leakage.

## Detection power

To assess the ability of the corrected source- and sensor-based estimators to detect system interactions, we used them as statistics for testing the null-hypothesis of no interaction (see ***Detection of system interactions***). We thus computed the sensitivity matrices of the test-statistics $T_{\text{imag}}^{(k)}$, $T_{\text{real}}^{(k)}$, and $T_{\text{compl}}^{(k)}$ corresponding to the source- and sensor-based estimators, where $k$ denotes the projection rank, which was set either to $k = 0$, corresponding to the uncorrected estimators, or to its maximal value $k = 21$. Detection power was quantified as

$$\frac{M + 1}{p(p - 1)/2},$$

where $M$ denotes the number of ordered interaction-pairs whose sensitivity is lower than the

sensitivity at the true interaction-pair $(k_0, l_0)$ and $p$ is the number of sensors. Detection power takes on values between $1/(p(p-1)/2 \approx 0.05$ and 1. If the sensitivity at the true interaction-pair is the highest of all $p(p-1)/2$ ordered source-pairs, $M = p(p-1)/2 - 1$ and detection power is 1 and if the sensitivity at the true source-pair is the lowest, $M = 0$ and detection power is $1/(p(p-1)/2)$. The significance-level for computing the sensitivity matrix was set to $\alpha = 0.025$ for the real and imaginary test-statistics and to $\alpha = 0.05$ for the complex test-statistic. Thus, when carrying out independent and uncorrected pairs-wise tests for significant interactions across all ordered pairs, the significant pairs (at the chosen significance level) correspond to the pairs with non-zero detection-power. Furthermore, the number of samples was set to $N = 100$, the noise-level to $\sigma = 0.01$, and the regularization-level to $\lambda = 10^{-2}$. The interaction lag was varied between 0 and 90 degrees. All other parameters were chosen as in Table 1.

Fig 6A (red curve) shows the detection power of the real part of the uncorrected source-based test-statistic as a function of lag. It is 0 for lags larger than about 65 degrees, which means that such interactions remain undetected. Between 0 and 60 degrees, its sensitivity gradually decreases. Furthermore, the detection power is at most about 0.8, which is rather low, for it means that 20% of the false positive interaction pairs have lower $p$-values than the true interaction pair. Fig 6B and 6C show that the real part of the uncorrected sensor-based test-statistic essentially behaves the same, although the detection power is higher (up to about 0.9). Fig 6D, 6E and 6F (red curves) show the detection power of the imaginary part of the uncorrected test-statistics. The red curves are hidden behind the green curves because the imaginary part of the test-statistics is invariant under the correction projection. Note that the imaginary parts only detect interactions with lags larger than about 40 degrees. However, for larger lags, the detection power is 1, hence the true interaction-pair has the smallest $p$-value of all significant interactions. Thus, for lags larger than 40 degrees, the imaginary part of both the source- and sensor-based test-statistics will detect the true interaction with probability $100 \times (1 - \alpha/2)$. Fig 6G, 6H and 6I (red curves) show the detection power of the complex-valued uncorrected test-statistics. Its sensitivity is high for all lags (at least 0.8), particularly for the sensor-based test-statistic. That is to say, the complex-valued test-statistic is *lag invariant*.

The detection power of the respective corrected test-statistics is also shown in Fig 6 (green curves). The real part has detection power 1 for lags up to about 50 degrees and is 0 for larger lags. The detection power of the imaginary part roughly behaves in the opposite way; it is 0 for lags up to about 30 degrees and is 1 for larger lags. The real and imaginary parts are hence complementary. The complex-valued test-statistics, on the other hand, have detection power 1 *for all lags*. We conclude that the corrected complex-valued test-statistics detect the true system interaction with probability $100 \times (1 - \alpha)$ irrespective of the lag.

## Reconstruction of functional networks

In the previous sections we have considered the case of two active and interacting sources. In practical applications, however, more than two sources might be active and interacting, thus forming a functional network. In this section we consider the detection power of the uncorrected and corrected estimators to detect interactions in functional networks and how detection power depends on the strength of the interactions. Since the difference in performance between the three correction methods is much smaller than that between the uncorrected and corrected estimators and because the sensor-based estimators performed slightly better than the source-based estimators, we restrict the analysis to the sensor-based estimator with correction in source-space and concentrate on the effect of the correction. We considered a network of $K$ evenly-spaced sources that are ordered from left to right in the one-dimensional source-space (see Section *Benchmark model*). The network structure is characterized by the spectral

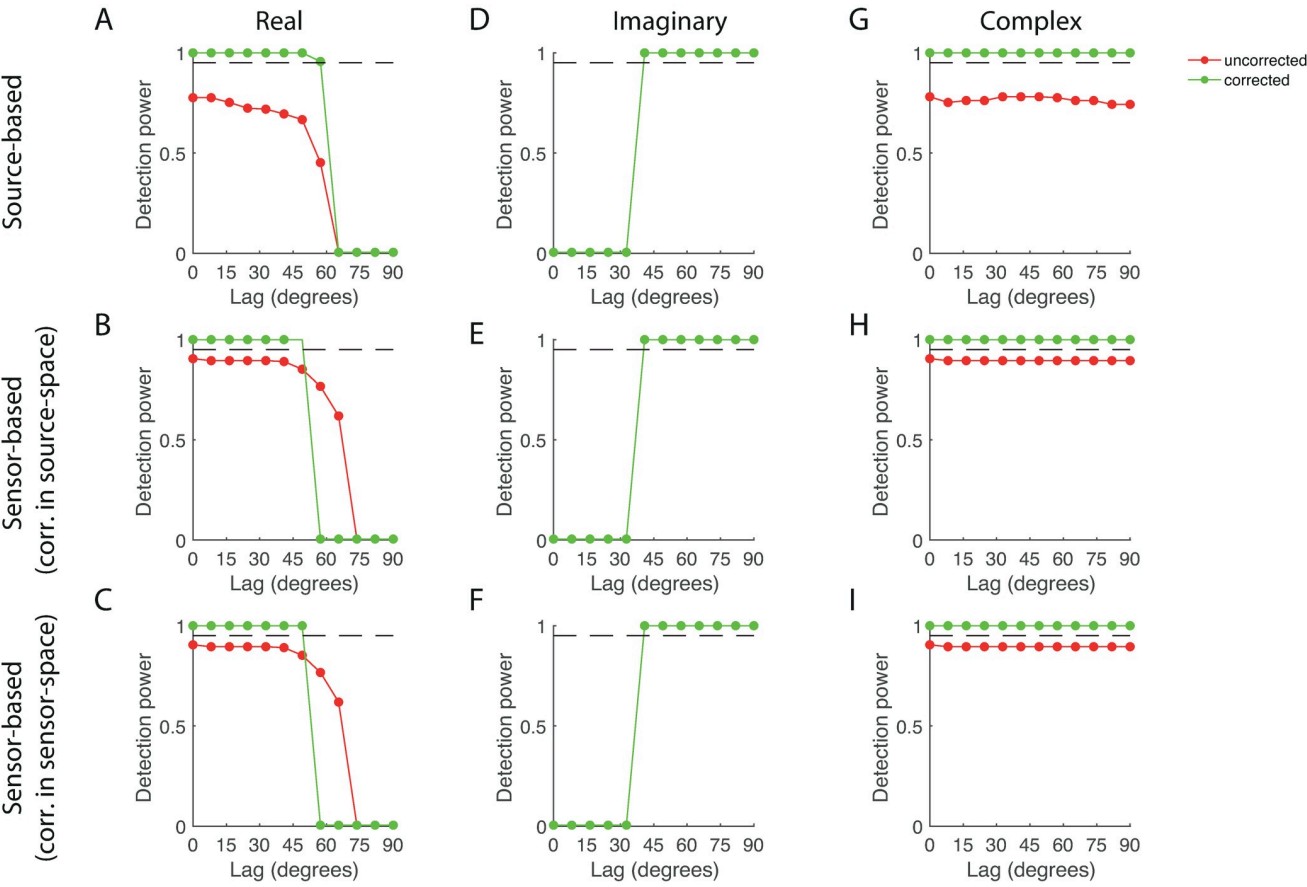

**Fig 6. Detection power.** A. Detection power of the real uncorrected (red) and corrected (green) source-based test-statistic. D. Detection power of the imaginary uncorrected (red) and corrected (green) source-based test-statistic. G. Detection power of the complex uncorrected (red) and corrected (green) source-based test-statistic. Panels B, E, and H: Same format as panels A, D, and G, respectively, but for the sensor-based test-statistic with correction in source-space. Panels C, F, and I: Same format as panels A, D, and G, respectively, but for the sensor-based test-statistic with correction in sensor-space.

matrix $vv^\dagger$, where $v \in \mathbb{C}^K$ is defined as

$$v = \sqrt{\gamma}\left(e^{i\phi_1}, e^{i\phi_2}, \cdots, e^{i\phi_K}\right)^T,$$

with $\phi_n = 2\pi(n-1)/K$, for $n = 1, \cdots, K$. The diagonal entries of the spectral matrix are set to 1 to that $\gamma$ is the shared coherence between the sources. Note that the phases $\phi_1, \cdots, \phi_K$ are evenly-spaced on the unit circle and that $\phi_1 = 0$ and $\phi_K = 2\pi - \phi_2$ and hence model a traveling wave of activity that propagates from left to right and has wavelength equal to the distance between the first and the last source. This model thus comprises a fully-connected network with $K$ different delays that are spatially organized as a propagating wave. The number of sources was set to $K = 4$. The network structure is shown in Fig 7A, 7B and 7C, which display, respectively, the real and imaginary parts and the absolute value of the spectral matrix. For this illustration $\gamma$ was set to 0.5. The absolute value of the spectral matrix (right panel) shows a fully-connected network of $K = 4$ sources. The $K = 4$, the delays between neighboring sources equals $\pi/4$. Thus, the real part of the spectral matrix (left panel) only shows the interaction between the first and the third source and between the second and the fourth source and the imaginary part of the spectral matrix (middle panel) only shows the interaction between all neighboring sources and, in addition, between the first and the third source. Thus, the real and

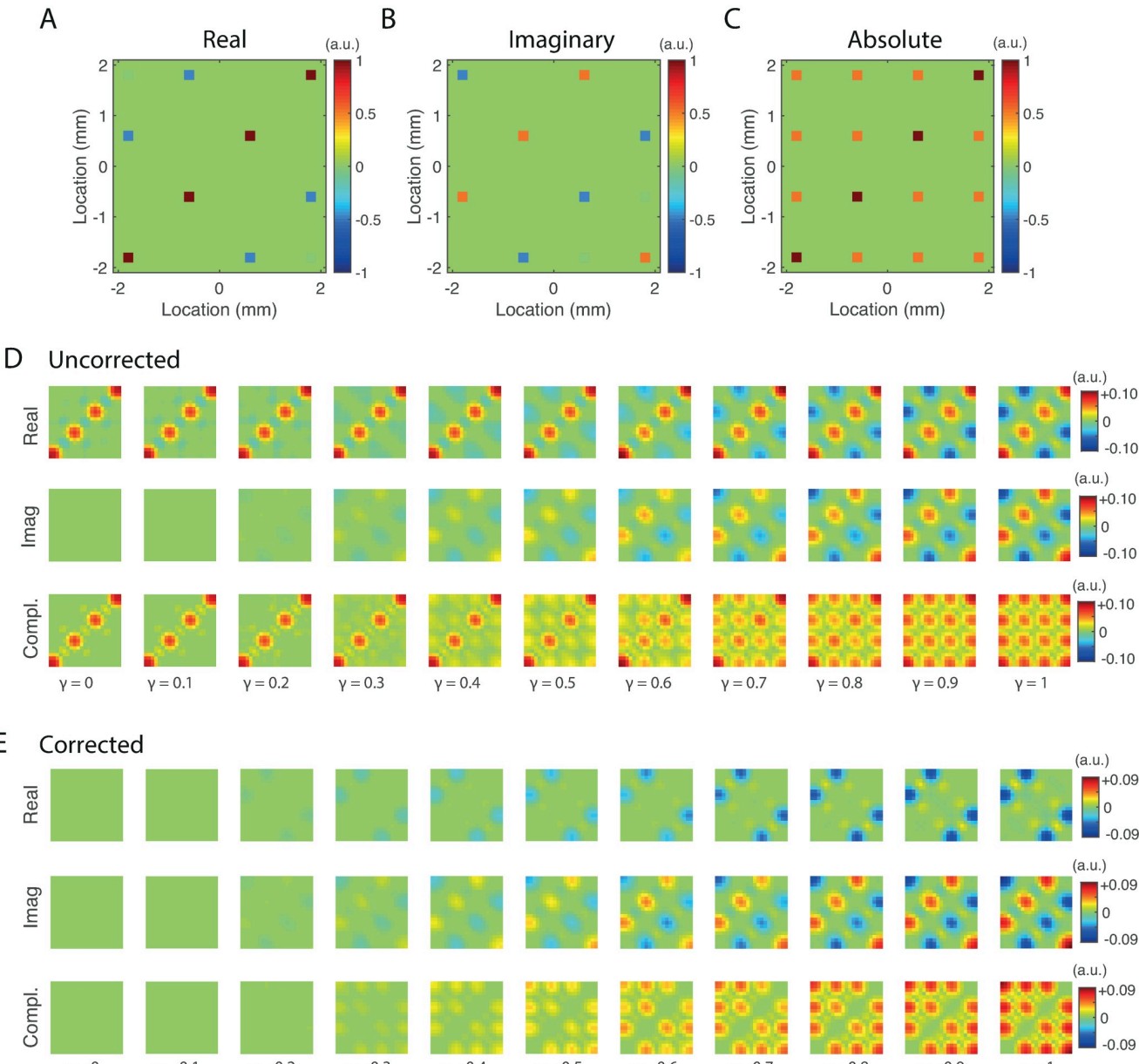

**Fig 7. Reconstruction of functional networks.** A. Real part of the true spectral matrix. B. Imaginary part of the true spectral matrix. C. Absolute value of the true spectral matrix. D. Sensitivity matrices of the real (top row), imaginary (middle row), and complex (bottom row) uncorrected sensor-based test-statistic as a function of interaction strength $\gamma$ ranging from 0 to 1 in steps in 0.1. E. Same as D. but for the corrected test-statistics.

imaginary parts "see" different interactions and only the absolute value shows the full network topology.

The number of samples was set to $N = 100$, the noise-level to $\sigma = 0.01$ and the regularization-level to $\lambda = 10^{-2}$. All other parameters, except the interaction parameters, were chosen as in Table 1. The interaction strength $\gamma$ ranged from 0 to 1 in steps of 0.1. A complete power analysis as carried out in the previous section involves conducting a separate analysis for each pair of sources, which is cumbersome and not very instructive. It is more illustrative to display the actual sensitivity matrices. The sensitivity matrices for the uncorrected test-statistics are

shown in Fig 7D. We make two observations. First, in the absence of interaction (i.e. $\gamma = 0$), the real and complex test-statistics have high sensitivity at and around the true source locations. When statistical tests are carried out this leads to false positives around the source locations. These spurious interactions have been referred to as *first-order* [37] and are due to the mere presence of active sources. No spurious interactions are present in the imaginary statistic because first-order spurious interactions are instantaneous. Second, when the interaction strength in increased, the sensitivity of all three test-statistics increases at and around the true interactions. Also the complex test-statistic, however, is sensitive to *both* real and imaginary interactions. The spurious interactions surrounding the true interactions have been referred to as *second-order* [37] or *ghost interactions* [38, 39]. They are independent of source-power and are due to the existence of true interactions. Note that all three test-statistics are equally affected by second-order spurious interactions. Incidentally, the local nature of second-order spurious interactions has been exploited in [39] to mitigate their effects. Fig 6E shows the sensitivity matrices of the corrected test-statistics. As in the previous sections, the projection rank was set to its maximal value of $k = 21$ so that the leakage term is completely suppressed. The figure shows that, in the absence of interaction, the sensitivity of the real and complex test-statistics is now zero hence no spurious interactions will be detected. Furthermore, when the interaction strength is increased, the sensitivity of all three test-statistic increases at and around the true interactions. Again, only the complex test-statistic is sensitive to both real and imaginary interactions. We conclude that, in this simple scenario, the corrected complex test-statistic is sensitive for all interactions, i.e. irrespective of their lag, and does not suffer from first-order spurious interactions. A price that is paid for the correction, however, are the artifacts in between the true source interactions, which appear as yellow dots (see right most panel in the top row of Fig 6E) and which are a consequence of the interaction term not being orthogonal to the leakage term (see ***The geometry of signal leakage***).

## Discussion and conclusions

### Summary and relevance

Detection of functional interactions in linearly-mixed systems is complicated by signal leakage, which refers to the incomplete unmixing of source signals due to the ill-posedness of the inverse problem under study [6]. If the forward model is (nearly) instantaneous, such as the forward models used for EEG/MEG [5] and local field potential (LFP) recordings [3], signal leakage is usually dealt with by discarding *all* instantaneous interactions, whether true or spurious. In practice, this is done by discarding the real part of complex-valued test-statistics and only retaining their imaginary part [11–15]. The drawback of this approach is that the sensitivity of the resulting test-statistics decreases with decreasing interaction lag, leading to false negatives and distortions in the topology of reconstructed functional networks [2]. More recent approaches exploit the relationship between signal leakage and the algebraic structure of linear inverse operators [24, 25, 27]. In [27] signal leakage is suppressed by correcting the sensor-space spectral matrix and subsequently projecting it to source-space using a non-linear dipole scanning algorithm [28]. To be applicable to more general source configurations, however, the correction method needs to be combined with more general inverse methods. In the current study we showed how to combine the correction with an arbitrary linear inverse operator and proposed two alternative methods in which the order of correction and projection is swapped, leading to novel sensor- and source-based estimators of system interactions. We demonstrated that all three estimators enable lag-independent detection of system interactions. As such, our study contributes to the development of reconstruction methods for system interactions from

linearly-mixed observations and we expect applications in neuroscience and the physical sciences.

## The geometry of signal leakage

We have provided a general characterization of signal leakage in source-space as characterized by spectral matrices that applies to all linear inverse operators. In particular, (the expected value of) the sample estimator of the source-space spectral matrix can be decomposed into three parts that are contained in different linear subspaces of the vector space of complex-valued (Hermitian) matrices. The first and second parts only depend on source-power and interactions, respectively, and we have referred to the corresponding subspaces as the *leakage* and *interaction subspaces*, respectively. The third part only depends on projected measurement noise and we have referred to the corresponding subspace as the *noise subspace*. In [27] a similar characterization is described for the sample estimators of the sensor-space spectral matrix. The difference between the source- and sensor-based characterizations of signal leakage is that, in the first case, the leakage and interaction subspaces are defined in terms of the resolution operator associated with a given inverse operator, whereas in the second case, the resolution operator is replaced by the forward matrix. These source- and sensor-space characterizations of signal leakage formalize the notions used in [37–39] in the special case in which connectivity is defined in terms of spectral matrices. In particular, the leakage term formalizes the equivalent notions of "first-order interactions" [37] and "spurious interactions" [38, 39]. The interaction term can be further decomposed into a term corresponding to true connectivity and a term corresponding to a type of spurious interaction that it independent of source power and which has been referred to as "second-order" [37] or "ghost" [38, 39] interactions. The current study focused on suppressing first-order interactions, whereas [38, 39] proposed a method to "bundle" second-order interactions and thereby reducing the number of false positives. However, our characterization of second-order interactions in terms of tensor-products of point-spread functions might be helpful in bundling these interactions.

## Source- versus sensor-based estimation of system interactions

We have described how leakage can be suppressed by applying projection operators to estimated source- or sensor-space spectral matrices. The source-space spectral matrices themselves can be estimated in two ways. The most obvious way is to project the observed sensor data to source-space and subsequently estimate the source-space spectral matrix from the reconstructed source-activity. An alternative way is to first estimate the sensor-space spectral matrix and subsequently projecting it to source-space by an inverse operator obtained by vectorizing a bilinear forward model in matrix space. This is essentially what has been proposed in [27]. We provided some insight into this estimator by giving an alternative characterization as the minimizer of a regularized cost function in matrix space. We have derived the filtering coefficients of both estimators and established a natural correspondence, which we exploited in the evaluation of their performance. In particular, the first estimator regularized with level $\lambda$ can be associated with the second estimator regularized with level $\lambda^2$. Corresponding filtering coefficients remain somewhat different, however, and this is reflected in the second estimator having a slightly higher spatial resolution.

## Leakage suppression in source- and sensor-space

We have formulated and evaluated three methods for suppressing signal leakage in reconstructed system interactions, each of which involves the application of a suitable projection operator. The projection operator can be applied in source-space, to either the source- or

sensor-based estimator of the source-space spectral matrix, or it can be applied in sensor-space, the latter corresponds to the correction method proposed in [27]. The sensor-based corrected estimators were more efficient in suppressing spurious interactions than the source-based corrected estimator and correction in source-space was more efficient than correction in sensor-space (Fig 4). The reductions in bias due to the corrections were comparable for the three methods (Fig 5). We subsequently used the corrected estimators as statistics for testing for significant interactions. Application to the model system demonstrated that the corrected test-statistics were able to detect system interactions, irrespective of the lag, with higher power than the uncorrected test-statistics. The detection power was the same for all three corrected test-statistics (Fig 6). Lastly, we showed that the corrected, but not the uncorrected, test-statistics are able to reconstruct functional networks comprising more than two sources (Fig 7). Taken together, we conclude that, out of the three test-statistics, the sensor-based test-statistic with correction in sensor-space is to be preferred, although the differences are small. We thus advice to use this test-statistic in practical applications to EEG/MEG and LFP recordings.

## Scope and limitations

This study focussed on theoretical aspects of reconstructing system interactions in linearly-mixed systems and did not address practical aspects. Two of these aspects are the selection of the regularization-level of the inverse operator and the rank of the correction operators. With regard to selecting appropriate values for the regularization parameters, several techniques can be explored including cross-validation [40], $L$-curve methods [41], residual analysis [42], and empirical Bayes [43]. Which method is most suited for which type of system and forward model is largely an empirical question that can be addressed through realistic numerical simulations. An interesting observation in this context is that the appropriate level of regularization depends on which aspect of the system under study is of interest. Thus, in [35] it was observed that estimation of brain interactions from MEG data requires less strong regularization than the reconstruction of the activity time-courses. In [36] this observation is formally investigated. Selection of the optimal rank of the projection operators remains an open question that can potentially yield interaction estimators with less bias and more powerful statistical hypothesis tests. In [27], the optimal projection rank was chosen on the basis of simulated data with a specific source configuration and parameter settings. For example, two sources were assumed to be active with equal strengths. However, no evidence was given to show that the obtained projection ranks are (close to) optimal for other source configurations and a data-driven selection method would therefore be preferable. The difficulty lies in the fact that the leakage, interaction, and noise terms are not separately observable, but only their superposition is. A data-driven method might be based on the sequence of estimates of the interaction matrices.

To determine to what extent the proposed estimators can successfully be applied to experimental EEG/MEG and LFP data requires characterizing and quantifying the effects of different factors that are present in such recordings such as cortical background activity, measurement noise, and inaccuracies in forward modeling, such as sensor positions, dipole orientations, and conduction properties of the volume conductor. Furthermore, although in this study we have considered functional networks comprising several sources, simulations of larger functional networks are required to obtain a better understanding of which types of network topologies can be successfully reconstructed and which ones are more challenging. Realistic simulations that take into account all of the above effects, however, cannot currently be carried out by standard implementations of the estimators due to its high memory demands. For example, implementation of the source-based estimator for a source-space of $p = 10^4$ voxels requires computing the singular value decomposition of the matrix $B_j$, whose dimensions are $p^2 \times p$,

which is not infeasible with standard algorithms. Similar considerations apply to the sensor-based estimator. In [27], these computational problems were circumvented by down-sampling the human cortex by a factor ten. Although this approach is suitable when using adaptive spatial filters for the inverse modeling, they are not appropriate when working with non-adaptive spatial filters as used in the current study, because such filters are unable to actively suppress source-activity from undesired locations. Thus, a natural follow-up study would be to use iterative reconstruction methods in combination with realistic EEG/MEG volume-conductor models and simulated functional networks to characterize the performance of the proposed methods.

## Supporting information

**S1 File.**
(M)

**S2 File.**
(M)

**S3 File.**
(M)

**S4 File.**
(M)

**S5 File.**
(M)

**S6 File.**
(M)

**S7 File.**
(M)

**S8 File.**
(M)

**S9 File.**
(M)

**S10 File.**
(M)

**S11 File.**
(M)

**S12 File.**
(M)

**S13 File.**
(M)

## Acknowledgments

The author would like to thank Wessel van Wieringen for valuable comments on the manuscript.

## Author Contributions

**Conceptualization:** Rikkert Hindriks.

**Formal analysis:** Rikkert Hindriks.

**Investigation:** Rikkert Hindriks.

**Methodology:** Rikkert Hindriks.

**Software:** Rikkert Hindriks.

**Writing – original draft:** Rikkert Hindriks.

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
