## [Decision Letter · Decision Letter 0]

19 Jun 2020

PONE-D-20-10601

Lag-invariant detection of interactions in spatially-extended systems using linear inverse modeling

PLOS ONE

Dear Dr. Hindriks,

Thank you for submitting your manuscript to PLOS ONE. After careful consideration, we feel that it has merit but does not fully meet PLOS ONE’s publication criteria as it currently stands. Therefore, we invite you to submit a revised version of the manuscript that addresses the points raised during the review process.

The reviewer raised important issues to the presentation and evaluation of the whole methodology.

I recommend to answer to all his comments one by one and re-submit the revised manuscript.

Important issues are the description of the formulas in a clear fashion to everyone and also to

submit in a public database your code for further validation from other researchers.

We look forward to receiving your revised manuscript.

Kind regards,

Stavros I. Dimitriadis

Academic Editor

PLOS ONE

Additional Editor Comments:

The reviewer raised important issues to the presentation and evaluation of the whole methodology.

I recommend to answer to all his comments one by one and re-submit the revised manuscript.

Journal Requirements:

Reviewers' comments:

Reviewer's Responses to Questions

**Comments to the Author**

1. Is the manuscript technically sound, and do the data support the conclusions?

Reviewer #1: Partly

2. Has the statistical analysis been performed appropriately and rigorously? 

Reviewer #1: Yes

3. Have the authors made all data underlying the findings in their manuscript fully available?

Reviewer #1: Yes

4. Is the manuscript presented in an intelligible fashion and written in standard English?

Reviewer #1: Yes

5. Review Comments to the Author

Reviewer #1: In this paper, the author proposes a novel method to address the important problem of estimating phase-synchronization while accounting for spurious interactions due to MEG field spread or EEG volume conduction. The method is based on a subspace projection to suppress the contribution of source power to the sensor-space spectral matrix. In contrast to the original paper proposing this approach (Ossadtchi et al. (2018)), this paper demonstrates how it can be combined with any linear inverse operator to obtain a lag-independent estimator of system interactions. The method is novel and promising and the paper is well-written. However, a number of concerns should be resolved before the paper could be considered for publication. Please see attached for details.

6. PLOS authors have the option to publish the peer review history of their article (what does this mean?). If published, this will include your full peer review and any attached files.

Reviewer #1: Yes: Nitin Williams

---

## [Author Response · Author response to Decision Letter 0]

21 Sep 2020

Below, the reviewer’s comments are displayed in black, authors responses in blue, and citations from the revised manuscript in green. 

In this paper, the author proposes a novel method to address the important problem of estimating phase-synchronization while accounting for spurious interactions due to MEG field spread or EEG volume conduction. The method is based on a subspace projection to suppress the contribution of source power to the sensor-space spectral matrix. In contrast to the original paper proposing this approach (Ossadtchi et al. (2018)), this paper demonstrates how it can be combined with any linear inverse operator to obtain a lag-independent estimator of system interactions. The method is novel and promising and the paper is well-written. However, a number of concerns should be resolved before the paper could be considered for publication: 

Major concerns: 

1. The method separates signal leakage-related and interaction-related contributions to sensor-space spectral matrix. However, it has been shown that the effect of signal leakage depends on the underlying true interaction strength (Fig. 3 in Palva et al. (2018)). In that sense, there is a portion of variance in the sensor-space spectral matrix that cannot be attributed to mere signal leakage or mere interaction but is due to the combination of the signal-leakage and underlying interaction strength. Does the method account for this?

For example, consider interactions between two different pairs of signals. The source-power for both pairs of signals are the same, while source-coherence for the two pairs is different. Due to the differences in source-coherence, the effect of ‘signal leakage’ will also be different. Now, would this difference be reflected in the ‘signal leakage’ term? Alternatively, does this constitute a case where the ‘source power’ and ‘source interactions’ do not lie in different linear subspaces, as the method assumes? 

When the interaction-strength is varied while keeping source-variance fixed (i.e. when source-coherence is varied), the corrected leakage term remains unchanged and the interaction term is scaled by the same factor. That is so say, the effect of source-coherence is a different relative weighting of the corrected leakage and interaction terms. I have added a formal description of this effect to the manuscript (see Section Suppression in source-space).

On a related note, I did not quite understand how the first term in equation 5 can have non-zero non-diagonal terms when in the previous line, it is mentioned the M can be split into M+ and M-, where M+ has all off-diagonal terms set to zero and M- has all diagonal terms set to zero. Given M+ has all off-diagonal terms set to zero, would we not expect the first term of equation 5 to also have all off-diagonal terms set to zero? 

In matrixform (i.e. non-vectorized form), the first term of Equation (5) is given by R*S^+*R^T, with R the resolution matrix. Although S^+ is indeed a diagonal matrix, R*S^+*R^T not necessarily is (the product of a diagonal matrix with two non-diagonal matrices is not necessarily a diagonal matrix). In fact, the non-diagonal entries of R*S^+*R^T correspond to first-order leakage. 

2. Related to point 1 above, how does the method’s performance compare to obtaining corrected interaction strength by subtracting estimated interaction strength from interaction strength obtained from a surrogate dataset? The surrogate dataset is such that it has also been subject to MEG field spread, its ‘source power’ is the same as the original dataset but ‘source interaction’ is set to zero. Would the correction by the proposed method be essentially equivalent to this? It would be important to compare the method’s performance with correction based on such a surrogate dataset. 

The reviewer raises an interesting point here. If the source power were known it could in principle be used to remove first-order leakage without suppressing second-order leakage. I tried this approach and I found that the estimated source power is much lower than the true power, which is because the power is smeared out over many locations due to the squared-norm penalty in the penalized least squares problem that is used to obtained the inverse operators. As a consequence, the correction has practically no effect. Because of this negative result I decided to not include it in the manuscript. 

3. The simulations based on two nodes are instructive and provide a basic demonstration of the method. However, since the assessment of the method is exclusively based on simulations, additional simulations should be done with more realistic simulated data. In particular, the correction should be applied to infer functional networks with multiple nodes. Between 80 and 120 nodes would make the network similar to those often estimated in MEG. The ‘true’ network should have interaction strengths ranging from 0 to 1 and the ability of the method to recover these strengths should be assessed. The amplitude and spectral properties of time series should also be similar to MEG data. For this, models of region dynamics such as Neural Mass Models (NMMs) (Jansen & Rit (1995)) could be used. Since properties of data and dimensionality of such a model would be similar to that encountered with MEG, good performance of the method on such a dataset would recommend its application to MEG. 

The reviewer raises an important point here. In the revised manuscript I have added a section named Reconstruction of functional networks in which the performance of the methods is assessed in reconstructing functional networks comprising several sources. Network dynamics is designed so that the interaction latencies cover the entire circle [0,360] degrees. Performance was assessed for interaction strengths ranging between 0 and 1.

I would like to clarify that the goal of the simulations is not to see how well the methods will perform on experimental EEG/MEG data, but to introduce the methods and to provide a proof-of-principle. I agree with the reviewer that realistic simulations need to be performed in order to assess the performance on experimental EEG/MEG data. Besides using a different forward model, this requires the inclusion of many more aspects such as cortical background activity, sensor noise, inaccuracies in the forward modeling, etc. Furthermore, the methods are computationally too expensive to be implemented on a high-resolution source-space. I discuss these issues in the discussion section of the revised manuscript (see the last paragraph): 

To determine to what extent the proposed estimators can successfully be applied to experimental EEG/MEG and LFP data requires characterizing and quantifying the effects of different factors that are present in such recordings such as cortical background activity, measurement noise, and inaccuracies in forward modeling, such as sensor positions, dipole orientations, and conduction properties of the volume conductor. Furthermore, although in this study we have considered functional networks comprising several sources, simulations of larger functional networks are required to obtain a better understanding of which types of network topologies can be successfully reconstructed and which ones are more challenging. Realistic simulations that take into account all of the above effects, however, cannot currently be carried out by standard implementations of the estimators due to its high memory demands. For example, implementation of the source-based estimator for a source-space of p = 10^4 voxels requires computing the singular value decomposition of the matrix B_j, whose dimensions are p^2\\times p, which is not infeasible with standard algorithms. Similar considerations apply to the sensor-based estimator. In [Ossadtchi, 2018], these computational problems were circumvented by down-sampling the human cortex by a factor ten. Although this approach is suitable when using adaptive spatial filters for the inverse modeling, they are not appropriate when working with non-adaptive spatial filters as used in the current study, because such filters are unable to actively suppress source-activity from undesired locations. Thus, a natural follow-up study would be to use iterative reconstruction methods in combination with realistic EEG/MEG volume-conductor models and simulated functional networks to characterize the performance of the proposed methods.

The use of neural mass models makes it more difficult to assess the performance of the methods proper and to disentangle it from the effects of pre-processing (e.g. bandpass filtering). This is because what effectively matters for performance is the distribution of the time-frequency coefficients of the sensor signals. Thus, by directly using time-frequency coefficients, performance can be disentangled from indirect issues. In fact, the time-frequency coefficients of linearized neural mass models with Gaussian white-noise input are independent and normally-distributed, just as in the manuscript, so that, as far as performance is concerned, the results are the same. I do acknowledge, however, that such models can be appropriate for testing entire processing pipelines as they are used in practice.

4. The paper describes three variants of the proposed method. For a user of the method, are there any technical reasons for choosing one variant over another (apart from simulation results)? Does one method perform better than others in all cases? If not, in what situations should one use one variant over another? Clear description of technical differences between variants and guidance on when to use one variant over another would be useful. Alternatively, the paper could focus on a single variant if one variant performs best in all cases or if there are only small differences in performance between the variants. 

For completeness, I prefer to keep all three methods. I added a paragraph in the discussion section I which I compare the performance of the three methods and draw a conclusion about which one to use in practice: 

The sensor-based corrected estimators were more efficient in suppressing spurious interactions than the source-based corrected estimator and correction in source-space was more efficient than correction in sensor-space (Figure 4). The reductions in bias due to the corrections were comparable for the three methods (Figure 5). We subsequently used the corrected estimators as statistics for testing for significant interactions. Application to the model system demonstrated that the corrected test-statistics were able to detect system interactions, irrespective of the lag, with higher power than the uncorrected test-statistics. The detection power was the same for all three corrected test-statistics (Figure 6). Lastly, we showed that the corrected, but not the uncorrected, test-statistics are able to reconstruct functional networks comprising more than two sources (Figure 7). Taken together, we conclude that, out of the three test-statistics, the sensor-based test-statistic with correction in sensor-space is to be preferred, although the differences are small. We thus advice to use this test-statistic in practical applications to EEG/MEG and LFP recordings.

5. All simulations were done with single examples. Where applicable, it would be good to perform simulations with multiple examples so that variability of method’s performance can be assessed. In this regard, particularly for Figs 3-5, it would be good to see confidence intervals marked for all line plots. 

Sample variability is assessed in the sections Detection Power and Reconstruction of functional networks. In these sections, the statistical power of the different test-statistics is approximated using Monte Carlo simulations. Specifically, the calculation of each point (on each curve) in Figure 6 of the revised manuscript (which corresponds to Figure 5 of the unrevised manuscript) is based on a large number of realizations of the generative model (10^3) that are used to approximate the statistics’ sample distributions and confidence regions (see Section Detection of system interactions). Thus, confidence regions are already incorporated in calculating these curves. Sections Effectiveness and Bias reduction cover performance aspects that have no inherent variability but relate to asymptotic properties of the estimators (effectiveness and bias). 

6. Early in the paper, it would be good to have a simple diagram illustrating the method’s operation, to complement the mathematical description. 

Thanks for the suggestion: This will indeed be helpful to the reader. I added a section named Subspace projections for suppression of spurious interactions that contains a figure illustrating the three different correction methods (see Figure 3 of the revised manuscript). 

7. In the Discussion, it would be good to describe any limitations of the method and particularly, in what practical situations might assumptions of the method be violated. Under what situations might errors be introduced into the estimation of corrected interaction strength (for e.g. low SNR? low lags?) How sensitive is the method to errors in the forward model? Further, how might the method work differently for EEG and for MEG data, and why? Some mention of these points is made while describing the method, but a fuller treatment in Discussion would be good. Any intuition on strengths, assumptions and limitations of the method would be very useful to future users.

The reviewer raises important questions relating to the application of the proposed methods to experimental EEG/MEG data. The manuscript focuses on the formulation and basic properties of the methods and I acknowledge that in order to be applicable to experimental EEG/MEG data, several questions needs to be addressed. These questions are ideally addressed in a separate study that entirely focuses on the application of the method to EEG/MEG data. In the discussion section I added a paragraph in which I discuss these issues (see cited text in response to comment 3.

8. A MATLAB and/or Python implementation of the proposed method could be made available to the community, in keeping with principles of Open Science. 

I fully agree and I will upload the documented Matlab code to a public repository. 

Minor concerns: 

1.Some typographical errors should be corrected: Page 3, final paragraph: “varies applications” should be “various applications” Page 10, final paragraph: “die to the mixing” should be “due to the mixing” 

Corrected. 

2.I might be mistaken, but in Page 17, would it not be that: T_imag=Imag(gamma(k,l))=abs(gamma(k,l))sin(theta(k,l)) , i.e. rather cos and T_real=Real(gamma(k,l))=abs(gamma(k,l))cos(theta(k,l)), i.e. rather than sin 

Yes indeed. Thanks for spotting these errors. 

3.Fig. 1B - z-axis labels should be specified 

Done. I have referred to the values on the color-axis as “arbitrary units” (a.u.) because I have set the tissue conductivity in Poisson’s equation to the arbitrary value of 1. 

4.Fig. 2A - colour of lines for ‘real’ and ‘imag’ too similar, make colour of lines different and include legend

Fig 2B - line colours and marker types too similar, change line colours and include legend

Fig 2C - white dots showing true interaction could be change to bigger black dots for visibility, z-axis labels should be specified Fig. 2 caption - “strength of interaction wat set” should be “strength of interaction was set” 

Done. 

5.Fig. 3 - should include legend for black lines in A, B and C! 

Done. 

6.Fig. 4 - white dots showing true interactions could be bigger black dots for visibility, z-axis labels of images should be specified, colorbar scales could be equal to aid comparisons 

Done. I left scaling of the color bars as it was because the overall scale is unimportant for performance: It’s the relative strength, i.e. across points in source-space, that determines performance and this is better visible now. 

7.Fig. 5 - x-axis labels could be 10 to 90, since 90 degrees is often mentioned in text. Panels A,C,E should be labelled ‘Source-based’ and changed to A,B,C. Panels B, D, F should be labelled ‘Sensor-based’ and changed to D,E,F. 

Done.

---

## [Decision Letter · Decision Letter 1]

9 Nov 2020

Lag-invariant detection of interactions in spatially-extended systems using linear inverse modeling

PONE-D-20-10601R1

Dear Dr. Hindriks,

We’re pleased to inform you that your manuscript has been judged scientifically suitable for publication and will be formally accepted for publication once it meets all outstanding technical requirements.

Kind regards,

Stavros I. Dimitriadis

Academic Editor

PLOS ONE

Additional Editor Comments (optional):

Dear Author

After carefully reading the revised manuscript and your response to the reviewer's comments,

I am glad to announce you that your paper has been accepted for publication in its

current form.

Reviewers' comments:

Reviewer's Responses to Questions

**Comments to the Author**

1. If the authors have adequately addressed your comments raised in a previous round of review and you feel that this manuscript is now acceptable for publication, you may indicate that here to bypass the “Comments to the Author” section, enter your conflict of interest statement in the “Confidential to Editor” section, and submit your "Accept" recommendation.

Reviewer #1: All comments have been addressed

2. Is the manuscript technically sound, and do the data support the conclusions?

Reviewer #1: Yes

3. Has the statistical analysis been performed appropriately and rigorously? 

Reviewer #1: Yes

4. Have the authors made all data underlying the findings in their manuscript fully available?

Reviewer #1: No

5. Is the manuscript presented in an intelligible fashion and written in standard English?

Reviewer #1: Yes

6. Review Comments to the Author

Reviewer #1: I thank the author for addressing my comments and commend the author on methodological development which is a valuable contribution to the field. A couple of typographical errors I spotted:

1. Figure 5, caption: In panels B, C, 507E, F, and G, the true interaction-pair is designated by the "white-circles." Should be "black circles"

2. Discussion, final paragraph: "not infeasible with standard algorithms". Should this be "infeasible with standard algorithms"?

7. PLOS authors have the option to publish the peer review history of their article (what does this mean?). If published, this will include your full peer review and any attached files.

Reviewer #1: **Yes: **Nitin Williams

---

## [Editor Report · Acceptance letter]

24 Nov 2020

PONE-D-20-10601R1 

Lag-invariant detection of interactions in spatially-extended systems using linear inverse modeling  

Dear Dr. Hindriks:

I'm pleased to inform you that your manuscript has been deemed suitable for publication in PLOS ONE. Congratulations! Your manuscript is now with our production department. 

Kind regards, 

on behalf of

Dr. Stavros I. Dimitriadis 

Academic Editor

PLOS ONE